# Summertime and wintertime atmospheric processes of secondary aerosol in Beijing

Jing Duan[1,2,3], Ru-Jin Huang[1,2], Yongjie Li[4], Qi Chen[5], Yan Zheng[5], Yang Chen[6], Chunshui Lin[1,2], Haiyan Ni[1,2], Meng Wang[1,2], Jurgita Ovadnevaite[7], Darius Ceburnis[7], Chunying Chen[8], Douglas R. Worsnop[9], Thorsten Hoffmann[10], Colin O'Dowd[7], Junji Cao[1,2]

[1]State Key Laboratory of Loess and Quaternary Geology (SKLLQG) and Key Laboratory of Aerosol Chemistry & Physics (KLACP), Institute of Earth Environment, Chinese Academy of Sciences, Xi'an 710061, China

[2]CAS Center for Excellence in Quaternary Science and Global Change, Chinese Academy of Sciences, Xi'an 710061, China

[3]University of Chinese Academy of Sciences, Beijing 100049, China

[4]Department of Civil and Environmental Engineering, Faculty of Science and Technology, University of Macau, Taipa, Macau 999078, China

[5]State Key Joint Laboratory of Environmental Simulation and Pollution Control, College of Environmental Sciences and Engineering, Peking University, Beijing 100871, China

[6]Chongqing Institute of Green and Intelligent Technology, Chinese Academy of Sciences, Chongqing 400714, China

[7]School of Physics and Centre for Climate and Air Pollution Studies, Ryan Institute, National University of Ireland Galway, University Road, Galway, H91CF50, Ireland

[8]CAS Key Laboratory for Biological Effects of Nanomaterials and Nanosafety, National Centre for 20   Nanoscience and Technology, Beijing 100191, China

[9]Aerodyne Research, Inc., Billerica, MA, USA

[10]Institute of Inorganic and Analytical Chemistry, Johannes Gutenberg University Mainz, Duesbergweg 10−14, Mainz 55128, Germany

*Correspondence to*: Ru-Jin Huang (rujin.huang@ieecas.cn) or Qi Chen (qichenpku@pku.edu.cn)

**Abstract**

Secondary aerosol constitutes a large fraction of fine particles in urban air of China. However, its formation mechanisms and atmospheric processes remain largely uncertain despite considerable studies in recent years. To elucidate the seasonal variations of fine particles composition and secondary aerosol formation, an Aerodyne quadrupole aerosol chemical 30   speciation monitor (Q-ACSM) combined with other online instruments were used to characterize the submicron particulate matter (diameter < 1 μm, $PM_1$) in Beijing during summer and winter 2015. Our results suggest that the photochemical oxidation was the major pathway for sulfate formation during summer, whereas aqueous-phase reaction became an important process for sulfate formation during winter. High concentration of nitrate (17% of the $PM_1$ mass) was found

during winter explained by enhanced gas-to-particle partitioning at low temperature, while high nitrate concentration (19%) was also observed under the conditions of high relative humidity (RH) during summer likely due to the hydrophilic property of $NH_4NO_3$ and hydrolysis of $N_2O_5$. As for organic aerosol (OA) sources, secondary OA (SOA) dominated the OA mass (74%) during summer while the SOA contribution decreased to 39% during winter due to enhanced primary emissions in the heating season. In terms of the SOA formation, photochemical oxidation perhaps played an important role for summertime oxygenated OA (OOA) formation and wintertime less oxidized OOA (LO-OOA) formation. The wintertime more oxidized OOA (MO-OOA) showed a good correlation with aerosol liquid water content (ALWC), indicating more important contribution of aqueous-phase processing than photochemical production to MO-OOA. Meanwhile, the dependence of LO-OOA and the mass ratio of LO-OOA to MO-OOA on atmospheric oxidative tracer (i.e., $O_x$) both degraded when RH were greater than 60%, suggesting that RH or aerosol liquid water may also affect the LO-OOA formation.

## 1. Introduction

Haze pollution in the North China Plain (NCP) occurs in all seasons and is characterized by high concentrations of fine particulate matter (PM) (Huang et al., 2014; Yang et al., 2015; An et al., 2019). The haze episodes have become more frequent and severe in recent years, although the annual average concentration of $PM_{2.5}$ (aerodynamic diameter ≤ 2.5 μm) in the NCP has decreased due to the implementation of a variety of regulatory measures (Fontes et al., 2017; Sun et al., 2016; Xu et al., 2018). A better understanding of the chemical nature, formation and transformation of fine particles is therefore essential for mitigating the haze pollution.

Compared to primary aerosol that is relatively well constrained in terms of the emission sources, secondary aerosol is still not well understood, likely due to its variable precursors, complex formation and atmospheric transformation processes mediated by meteorological conditions (Ding et al., 2016; Petäjä et al., 2016; Tie et al., 2017; Xu et al., 2017). A number of studies have investigated the formation and aging processes of secondary aerosol (Takegawa et al., 2009; Huang et al., 2014, 2019; Sun et al., 2014, 2016; Cheng et al., 2016; Wang et al., 2016; Duan et al., 2019). For example, field measurements showed that aqueous-phase oxidation of $SO_2$ could be an important formation pathway of sulfate at high RH during haze events (Sun et al., 2014; Elser et al., 2016). Our recent study, however, suggested that the role of photochemical oxidation versus aqueous-phase reactions in the sulfate formation largely depends on the meteorological conditions (Duan et al., 2019). Moreover, the aqueous oxidation of $SO_2$ by $NO_2$ has been suggested to be an efficient pathway for sulfate formation (Cheng et al., 2016; Wang et al., 2016), although contribution from this pathway in real air is controversial and is still an open question (Guo et al., 2017; Liu et al., 2017). Model simulation has showed that heterogeneous sulfate formation from $SO_2$ oxidation catalyzed by $Fe^{3+}$ in aerosol water can significantly improve the sulfate simulation (Li et al., 2017). These studies show that even for sulfate, a simple inorganic

component, the formation is complex during haze events. As for secondary organic aerosol (SOA), the formation pathways are much less understood given the complexity in SOA composition and its precursors. The contribution of SOA to $PM_{2.5}$ mass was found to be as important as secondary inorganic aerosol (SIA-sulfate, nitrate and ammonium) during severe haze events (Huang et al., 2014). Positive matrix factorization (PMF) analyses have resolved multiple OA factors. In particular, photochemical oxidation has been suggested to be the major pathway of SOA formation during some pollution events in some cities (e.g., Beijing and Xi'an) because SOA correlated tightly with odd oxygen ($O_x$) and was independent of RH (e.g., Sun et al., 2014; Elser et al., 2016). However, aqueous-phase formation of SOA has also been considered as an important pathway during some pollution events in e.g., Beijing, Shijiazhuang and Baoji (Sun et al., 2016; Wang et al., 2017; Huang et al., 2019). The formation of less oxidized oxygenated OA (LO-OOA) and more oxidized oxygenated OA(MO-OOA) in Baoji seemed being significantly influenced by aqueous-phase chemistry during the period of low atmospheric oxidative capacity (Wang et al., 2017), while in another study Xu et al. (2017) suggested that aqueous-phase processing has a dominant impact on the MO-OOA formation but photochemical oxidation is the dominant pathway for the LO-OOA formation in Beijing. It is unclear yet whether the aqueous-phase processing plays a key role in the haze development and what the mechanisms are.

Based on previous studies, it is evident that the formation of secondary aerosol shows large spatial and temporal variations and may vary in different events especially for the SOA formation. Further studies are therefore needed to better understand the formation and transformation of secondary aerosol in polluted air. As the pollutant emissions, meteorological conditions and oxidation capacity vary dramatically between summer and winter, in order to further elucidate the atmospheric processes of secondary aerosol under distinctly different atmospheric conditions, here we present results from summertime and wintertime measurements in 2015 in Beijing. The seasonal difference in aerosol chemical composition and in formation mechanisms are discussed.

## 2. Experimental

### 2.1 Measurement site

The summer (1 July to 19 August 2015) and winter (4 December 2015 to 6 February 2016) campaigns were conducted at an urban site on the roof of a five-story building (~20 m above the ground) in the National Center for Nanoscience (39.99°N, 116.32°E) in Beijing, which is close to the fourth ring of Beijing and surrounded by residential, commercial and traffic areas.

### 2.2 Instrumentation

An Aerodyne quadrupole ACSM (Q-ACSM) with a time resolution of ~30 min was deployed for the continuous measurement of NR-$PM_1$ species including organics, sulfate, nitrate, ammonium and chloride. Detailed operation principles of ACSM can be found elsewhere (Ng et

al., 2011; Wang et al., 2017; Duan et al., 2019). Briefly, ambient air was sampled through a 3/8 inch stainless steel tube at a flow rate of ~3 L min$^{-1}$, and the coarse particles were removed by an Uuniversity Research Glassware (URG) cyclone (Model: URG-2000-30ED) with a size cut of 2.5 μm in front of the sampling inlet. A Nafion dryer (MD-110-48S, Perma Pure, Inc., Lakewood, NJ, USA) was applied to dry aerosol particles before entering the ACSM and the submicron aerosol was subsampled into the ACSM at a flow rate of 85 mL min$^{-1}$ through a 100 μm diameter critical aperture. The submicron particles were focused into a narrow beam by an aerodynamic lens and impacted a hot vaporizer (~600℃). The resulting vapor was ionized with electron impact (70 eV) and chemically characterized with a quadrupole mass spectrometer. Mono-dispersed 300 nm ammonium nitrate particles, generated by an atomizer (Model 9302, TSI Inc., Shoreview, MN, USA) and selected by a differential mobility analyzer (DMA, TSI model 3080), were used to determine the response factor (RF) and calibrate the ionization efficiency (IE) for the ACSM (Ng et al., 2011).

An Aethalometer (Model AE-33, Magee Scientific) was used for the determination of equivalent black carbon (eBC) concentration with a time resolution of 1 min. $SO_2$, CO, $NO_x$ and $O_3$ were measured by an Ecotech EC 9850 sulfur dioxide analyzer, a Thermo Scientific Model 48i carbon monoxide analyzer, a Thermo Scientific Model 42i $NO$-$NO_2$-$NO_x$ analyzer and a Thermo Scientific Model 49i ozone analyzer, respectively. Meteorological parameters, including wind speed, wind direction, RH, and temperature were measured by an automatic weather station (MAWS201, Vaisala, Vantaa, Finland) and a wind sensor (Vaisala Model QMW101-M2).

**2.3 Data analysis**

**2.3.1 ACSM data analysis**

The standard ACSM data analysis software in Igor Pro (WaveMetrics, Inc., Lake Oswego, Oregon USA) was used to analyze the ACSM dataset. IE was determined by comparing the response factors of ACSM to the mass calculated with the known particle size and the number concentration from a condensation particle counter (CPC, TSI model 3772). Standard relative ionization efficiencies (RIEs) were used for organics, nitrate and chloride (i.e., 1.4 for organics, 1.1 for nitrate and 1.3 for chloride) (Canagaratna et al., 2007) and RIEs for ammonium (6.1) and sulfate (1.2) were estimated from the RIE calibrations using $NH_4NO_3$ and $(NH_4)_2SO_4$. The collection efficiency (CE) was introduced to correct for the particle loss due to particle bounce, which is influenced by aerosol acidity, composition and the aerosol liquid water content. Considering that the particles were dried before entering the ACSM and are overall neutralized, the influences of aerosol liquid water and acidity are expected to be negligible. Therefore, CE was determined as $CE_{dry} = \max (0.45, 0.0833 + 0.9167 \times ANMF)$, where ANMF represents the mass fraction of ammonium nitrate in NR-$PM_1$ (Middlebrook et al., 2012).

**2.3.2 Source apportionment**

PMF was used to perform the source apportionment on the mass spectra of organics as

implemented by the multilinear engine (ME-2; Paatero, 1999) via the interface SoFi (Source Finder) coded in Igor Wavemetrics (Canonaco et al., 2013). The OA source apportionment for the summer dataset and the winter dataset was performed separately. First, a range of solutions with different factor numbers from unconstrained runs were examined using conventional PMF. A mixture of two or more factors was found and cannot be separated further for both summer and winter dataset even after increasing the factor number. Then the ME-2 approach was used as it can direct the apportionment towards an environmentally-meaningful solution by introducing *a priori* information (profiles) for certain factors (Canonaco et al., 2013; Crippa et al., 2014; Frohlich et al., 2015). The final results are verified based on the rationality of unconstrained factors, distinct mass spectra, time series and good correlations with external tracers for all factors. Details about the PMF and ME-2 source apportionment are provided in the supplementary information. In this study, four OA factors including hydrocarbon-like OA (HOA), cooking OA (COA), isoprene-oxidized OA (ISOOA) and OOA were resolved during summer, and six OA factors including HOA, COA, biomass burning OA (BBOA), coal combustion OA (CCOA), LO-OOA, and MO-OOA were resolved during winter. Note that MO-OOA and LO-OOA were defined in winter because of much higher $f_{44/43}$ value of MO-OOA (8.6) than that of LO-OOA (2.6) and the large difference in time series of these two factors. Different from the two identified wintertime OOAs, the summertime OOA was mixed and could not be further separated into LO-OOA and MO-OOA, likely due to the limited mass resolution of ACSM (Sun et al., 2012).

**2.4 Aerosol liquid water content**

Aerosol liquid water content (ALWC) was calculated by the ISORROPIA-II model (Fountoukis and Nenes, 2007) using ACSM aerosol composition and meteorological parameters (temperature and RH) as model inputs. The ISORROPIA-II model then calculated the composition and the phase state of a $NH_4^+$–$SO_4^{2-}$–$NO_3^-$–$Cl^-$–$H_2O$ system in thermodynamic equilibrium and the concentration of $H^+$ and ALWC can be resolved.

**3.  Results and discussion**

**3.1 Chemical composition of $PM_1$ in summer and winter**

Fig. 1 shows the time series of individual chemical composition of $PM_1$, OA sources, gaseous species and meteorological parameters in summer and winter. The average values of each components in different periods are summarized in Table 1. As shown in Fig. 1, very different time variations of $PM_1$ composition, OA sources and gaseous species were observed between summer and winter. The concentrations of gaseous species such as CO, $SO_2$ and $NO_x$ during winter were ~3 times higher than those during summer, largely due to enhanced emissions and accumulation of pollutants during winter. In contrast, the average $O_3$ concentration during winter (9.2 ppb) was ~5 times lower than that during summer (49.6 ppb), largely due to titration by

enhanced wintertime emissions of $NO_x$. In summer, meteorological conditions were relatively stable with the wind speeds often being lower than 2 m s$^{-1}$ during the measurement period. The average mass concentration of PM$_1$ was 41.0 ± 23.4 µg m$^{-3}$ during summer, and OA constituted a major fraction of PM$_1$ mass (47%), followed by sulfate (18%), ammonium (14%), nitrate (13%),

eBC (7%), and chloride (1%). In winter, however, due to frequent changes of meteorological conditions, the time series of PM$_1$ components and gaseous species varied dramatically, such as the rapid build-up of haze pollution under stagnant meteorological conditions with high RH and low wind speed (< 2 m s$^{-1}$), or sudden occurrence of clean days because of the dilution by clean air masses from north or northwest at high wind speed (5-10 m s$^{-1}$). The PM$_1$ mass concentration

with an average value of 63.2 ± 55.1 µg m$^{-3}$ was observed during winter. OA contributed a mass fraction of 49%, followed by nitrate (17%), sulfate (12%), ammonium (12%), eBC (5%), and chloride (5%). As for secondary inorganic species, sulfate is the largest contributor in summer, but replaced by nitrate in winter largely due to different formation processes between summer and winter (as discussed below). An enhancement of chloride from ∼1% (0.2 µg m$^{-3}$) in summer

to ∼5% (3.1 µg m$^{-3}$) in winter was also observed, which could be attributed to substantial emissions from coal combustion in winter (Huang et al., 2014; Wang et al., 2015; Elser et al., 2016; Hu et al., 2016).

   OA contributed the highest mass fraction to PM$_1$ in both summer and winter, suggesting the important role of OA in PM$_1$ pollution (Zhang et al., 2014; Sun et al., 2015; Hu et al., 2016). As for

OA sources, the concentration of HOA increased from 1.5 µg m$^{-3}$ during summer to 3.6 µg m$^{-3}$ during winter and the fractional contribution of HOA increased accordingly from 8% to 12%. COA also showed similar increasing trend from summer to winter, with mass concentration increasing from 3.4 µg m$^{-3}$ to 6.3 µg m$^{-3}$ and fractional contribution from 18% to 20%. The increase in mass concentration of COA in winter is likely caused by meteorological conditions (e.g., shallower PBL

heights in winter than in summer) because the cooking activities are not expected to have seasonal variation. BBOA and CCOA were only resolved during winter with mass fractions of 9% and 20%, respectively. The average wintertime BBOA concentration decreased from 3.6 µg m$^{-3}$ in 2010 (Hu et al., 2016) to 2.8 µg m$^{-3}$ in 2015 in our study and further to 2.4 µg m$^{-3}$ in 2017-2018 (Li et al., 2019), suggesting efficient control of the biomass burning activities in Beijing and

surrounding areas in recent years. Two secondary OA sources (OOA and ISOOA) were resolved during summer. OOA was the dominant OA source during summer due to strong atmospheric oxidation capacity, accounting on average for 69% of total OA with an average mass concentration of 13.0 µg m$^{-3}$. In addition to OOA, another SOA factor, ISOOA, derived from isoprene-oxidation was resolved in summertime Beijing. ISOOA was generally thought to be formed in environment

with low NOx and high biogenic emissions, but has recently been observed in urban Nanjing in summer 2013 (Zhang et al., 2017). Similar to the result in summertime Nanjing (4% of OA), in our study ISOOA is found to contribute ∼5% of total OA with an average mass concentration of 1.0 µg m$^{-3}$ in summertime Beijing. In contrast, ISOOA was not resolved during winter, which is consistent

with the low emissions of isoprene. It should be noted that the estimated uncertainty of ISOOA factor is ~20%, suggesting large uncertainty in ISOOA source in urban Beijing. In winter, two OOAs, i.e., LO-OOA and MO-OOA, are identified. The fractional contribution of LO-OOA to total OA (23%, 7.1 μg m$^{-3}$) was larger than that of MO-OOA (16%, 5.1 μg m$^{-3}$), suggesting more efficient formation of LO-OOA than MO-OOA during winter. The OOA contribution to total OA decreased from 69% in summer to 39% in winter, mainly due to enhanced primary emission in winter.

We also compared our results with previous studies in Beijing. As shown in Fig. 2, the PM$_1$ mass concentration during summer in our study is similar to that observed during summer 2012 (Hu et al., 2017), but ~30-50% lower than that measured during summer 2008 (Zhang et al., 2013) and summer 2011 (Sun et al., 2015; Hu et al., 2016). From summer 2008 to summer 2015, the fraction contribution of SIA to the PM$_1$ mass decreased from 62% to 45% and the OA contribution increased from 36% to 47%, suggesting the increasing importance of OA due to the reduction of SO$_2$ and NO$_x$ emissions. Meanwhile, we found that the fractional contribution of SOA (i.e., OOAs herein) to total OA also increased from 60% in summer 2008 to 69% in summer 2015, indicating the increased importance of SOA formation during summer in recent years. From summer 2015 to summer 2018, the PM$_1$ concentration continued decreasing while the SIA contribution was again higher than that during summer 2015 and the SOA faction was similar to that in summer 2015 (Zhou et al., 2019). In winter, the average concentration of PM$_1$ observed in our study was similar to that observed in 2014 (Xu et al., 2018), but lower than most of those in 2008-2013 (Zhang et al., 2013; Zhang et al., 2014; Sun et al., 2013, 2015, 2016; Hu et al., 2016). Especially in winter 2013 enhanced average PM$_1$ concentration of 94.0 μg m$^{-3}$ and 81.7 μg m$^{-3}$ were observed by Zhang et al (2014) and Hu et al (2017), respectively, due to the severe pollution events in January 2013. After 2015, the wintertime PM$_1$ concentration further decreased to 33.2 μg m$^{-3}$ in 2017-2018 (Li et al., 2019), but showed a peak concentration (92.9 μg m$^{-3}$) in Nov-Dec 2016 because of prevailing air masses from south and northeast of Beijing and high frequencies of high RH and low wind speeds (Xu et al., 2018). Such changes again suggested that the difference in meteorological conditions is one of the major causes of severe particulate pollution in Beijing. The variations of SIA fractions were not obvious from 2008 to 2015 during wintertime with SIA contribution ranged from 34% to 43%, except a higher SIA contribution of 55% observed by Hu et al (2017) in winter 2013. However, from winter 2015 to 2018, the SIA fraction increased from 40% to 54%, suggesting the increased importance of SIA from 2015 to 2018. As for OA, SOA have lower contribution during winter than summer in all years from 2008 to 2018. The SOA contributions after 2013 (40%-54%) were higher than most of those in 2008-2012 (20%-30%) during wintertime, highlighting the increased importance of SOA in winter.

## 3.2 Daytime evolution of secondary species in summer and winter

The diurnal cycles of PM$_1$ species and OA sources are shown in Fig. S4. The diurnal variations are affected by the evolution of planetary boundary layer (PBL) height that governs the vertical

dispersion of pollutants and by the diurnal cycle of the emissions and atmospheric processes (Huang et al., 2019). To minimize the effects of PBL height and to investigate the daytime processes of secondary species between summer and winter, data of secondary species were normalized by ΔCO because CO is often used as a tracer to account for dilution on timescales of hours to days. As shown in Fig. 3, after offsetting the PBL dilution effect, OOA displayed the fastest increase from 6:00 to 13:00 (local time, LT) with an average growth rate of 2.4 $\mu g\ m^{-3}\ ppm^{-1}\ h^{-1}$ during summer. Sulfate also showed a clear increasing trend from 6:00 to 17:00 LT with an average growth rate of 1.05 $\mu g\ m^{-3}\ ppm^{-1}\ h^{-1}$, indicating the efficient daytime production of OOA and sulfate. However, nitrate displayed a decreasing trend from 11:00 to 18:00 LT, suggesting that evaporation of existing ammonium nitrate was more efficient than production because of its high volatility and thermal instability in the warm afternoon. Associated with sulfate and nitrate, the diurnal cycle of ammonium was affected by both sulfate and nitrate formation mechanisms, which showed a minor increasing trend from 6:00 to 15:00 LT with an average growth rate of 0.48 $\mu g\ m^{-3}\ ppm^{-1}\ h^{-1}$. In contrast, all secondary species displayed similar increasing trends from 8:00 to 13:00 LT during winter. Nitrate showed the fastest increasing rate of 1.2 $\mu g\ m^{-3}\ ppm^{-1}\ h^{-1}$ on average, followed by LO-OOA (1.1 $\mu g\ m^{-3}\ ppm^{-1}\ h^{-1}$), ammonium (0.6 $\mu g\ m^{-3}\ ppm^{-1}\ h^{-1}$), sulfate (0.48 $\mu g\ m^{-3}\ ppm^{-1}\ h^{-1}$) and MO-OOA (0.19 $\mu g\ m^{-3}\ ppm^{-1}\ h^{-1}$). The high increasing rate of nitrate in winter compared to summer suggested its enhanced daytime production due to lower temperature. LO-OOA displayed a much clearer increasing trend than that of MO-OOA, suggesting that the daytime formation of LO-OOA was much more efficient than that of MO-OOA during winter, which is consistent with the diurnal cycle of MO-OOA concentration (Fig. S4). It should be noted that although the increase rates of sulfate and SOA (LO-OOA + MO-OOA) during winter were lower than that during summer, the enhancement ratios (about 1.5 times, from 4.0 $\mu g\ m^{-3}\ ppm^{-1}$ to 6.1 $\mu g\ m^{-3}\ ppm^{-1}$ for sulfate; and about 2.0 times, from 5.4 $\mu g\ m^{-3}\ ppm^{-1}$ to 11.4 $\mu g\ m^{-3}\ ppm^{-1}$ for SOA) during winter were similar with that during summer (about 2.0 times, from 10.3 $\mu g\ m^{-3}\ ppm^{-1}$ to 21.9 $\mu g\ m^{-3}\ ppm^{-1}$ for sulfate; and 1.8 times, from 20.7 $\mu g\ m^{-3}\ ppm^{-1}$ to 38.8 $\mu g\ m^{-3}\ ppm^{-1}$ for SOA), indicating that the daytime oxidation formation efficiency of secondary aerosol during winter was comparable to that during summer. Certainly, we note that wintertime photochemistry was traditionally thought to be low because the wintertime OH radical concentrations were traditionally thought to be low due to low $O_3$ concentrations (e.g., 9.2 ppb in winter versus 49.6 ppb in summer in our study) and reduced solar UV level in winter. However, recent measurements of OH radicals at a suburban site of Beijing reported wintertime OH radical concentrations at noontime ranged from 2.4×10⁶ $cm^{-3}$ during severe pollution events ($k_{OH}$ ~27 $s^{-1}$) to 3.6×10⁶ $cm^{-3}$ during relatively clean events ($k_{OH}$ ~5 $s^{-1}$) (Tan et al., 2018). These OH radical concentrations are nearly one order of magnitude larger than those predicted by global models for northern China in winter (Lelieveld et al., 2016). The higher-than-expected OH concentrations and moderate OH reactivity in Beijing resulted in fast photochemistry in winter (Lu et al., 2019), explaining the high contribution of secondary aerosol during wintertime severe haze events

(Huang et al., 2014). Meanwhile, there have been increasing evidence showing HONO, instead of $O_3$, is the main source of OH radicals in wintertime North China Plain (Hendrick et al., 2014; Xing et al., 2019). Further, Lu et al. (2019) found surprisingly high OH radical oxidation rates of up to 15 ppbv $h^{-1}$ in Beijing in winter, which is comparable to that in summer and is mainly initiated by the photolysis of HONO and maintained by an extremely efficient radical cycling process driven by nitric oxide.

**3.3 Sulfate and nitrate formation in summer and winter**

To elucidate the formation mechanisms of sulfate and nitrate, the correlations between $NO_3^-$ and $SO_4^{2-}$ colored by RH or $O_x$ are shown in Fig. 4. $NO_3^-$ and $SO_4^{2-}$ correlated well in winter with $R^2$ of 0.6, while there was no good correlation between $NO_3^-$ and $SO_4^{2-}$ in summer. The correlations between $NO_3^-$ and $SO_4^{2-}$ strongly depended on RH and $O_x$, and this dependence on RH and $O_x$ is reverse between summer and winter. In summer, the correlation between $NO_3^-$ and $SO_4^{2-}$ showed higher slope at higher RH, indicating a faster production rate of $NO_3^-$ than $SO_4^{2-}$ in high RH condition. The higher concentration of $NO_3^-$ at high RH was likely due to the transformation of $HNO_3$ into particle phase, which could be enhanced at high RH and low temperature conditions (Sun et al., 2015). In contrast, the slope of correlation between $NO_3^-$ and $SO_4^{2-}$ decreased as $O_x$ concentration increased, indicating the efficient formation of $SO_4^{2-}$ in high $O_x$ concentration due to photochemical oxidation. On the contrary, in winter both opposite trends related to RH and $O_x$ were observed. Higher $NO_3^-/SO_4^{2-}$ ratio was found at low RH and high $O_x$ conditions and the $NO_3^-/SO_4^{2-}$ ratio decreased dramatically as RH increased due to the fast formation of sulfate in high RH condition. Such variations between summer and winter suggested that in summer photochemical oxidation was perhaps the major pathway of sulfate formation and high RH condition facilitated the nitrate formation due to enhanced gas-to-particle partitioning of $NH_4NO_3$, while in winter daytime photochemical process played a dominant role in the formation of nitrate and aqueous-phase processes had significant impacts on the formation of sulfate.

This finding is further supported by the effects of RH and $O_x$ on sulfur oxidation ratio (SOR) and nitrogen oxidation ratio (NOR) (Fig. S5). In summer, there was no obvious dependence between SOR and RH, whereas a positive correlation was found between SOR and $O_x$. NOR was independent of both RH and $O_x$, the higher nitrate concentration in high RH condition during summer could be attributed to enhanced gas-to-particle partitioning of the highly hydrophilic $NH_4NO_3$ and the hydrolysis of $N_2O_5$ (Su et al., 2017). In contrast, SOR shows an evident exponential relationship with RH in winter, indicating the important role of aqueous-phase reaction in the sulfate formation. NOR shows just a small increasing trend with RH, but strongly correlated with $O_x$ concentration, suggesting the major role of photochemical oxidation in the nitrate formation during winter. It should be noted that SOR also displayed an increasing trend as $O_x$ increased during winter, implying the role of photochemical process in the sulfate formation and high RH condition still promoted the gas-to-particle partitioning of nitrate. However, as the aqueous-

phase reaction leads enhanced sulfate formation during winter, the ratio of $NO_3^-/SO_4^{2-}$ decreased as RH increased.

It is noted that the diurnal variations of $SO_4^{2-}/\triangle CO$ and $NO_3^-/\triangle CO$ were similar in winter (Fig. 3). However, the increasing rate of $SO_4^{2-}/\triangle CO$ was much lower than that of $NO_3^-/\triangle CO$. Also the absolute concentration of wintertime sulfate showed a decreasing trend in the afternoon, consistent with the diurnal variations of RH and ALWC, while the absolute concentration of wintertime nitrate showed an increasing trend from 8:00 to 18:00 LT (Fig. S4). These results further suggest the difference in dominant formation processes between sulfate and nitrate in winter. The photochemical processes were likely the dominant pathways for daytime nitrate formation but less important for sulfate formation because of large contributions from aqueous-phase processes as indicated by the exponential relationship between SOR and RH in wintertime Beijing (Fig. S5d).

**3.4 SOA formation in summer and winter**

The relationships between OOA and $O_x$ or ALWC during summer are shown in Fig. 5. It should be note that a majority of the data are scattering, therefore caution should be given for the uncertainties when using $O_x$ and ALWC as indicators of photochemical processing and aqueous processing, respectively. The mass fraction of OOA to OA ($f_{OOA}$) is positively correlated with $O_x$, increasing from 0.61 to 0.75 when $O_x$ concentration increases from 20 ppb to 150 ppb, suggesting that photochemical oxidation may play a key role in the OOA formation during summer in Beijing. In contrast, $f_{OOA}$ was independent of ALWC during summer. Consistently, the mass concentration of OOA was also positively correlated with $O_x$ concentration, whereas higher RH was usually located at the left-bottom corner with low $O_x$ concentration and low OOA concentration (Fig. 5c). This suggests that RH has relatively minor but complex effects on the formation of OOA during summer. During winter, although the $f_{OOA}$ (mass fraction of LO-OOA + MO-OOA to total OA) showed no clear increase or decrease trends with $O_x$ or ALWC (Fig.5 e, f), the correlation between the mass concentration of OOA (total mass of LO-OOA+MO-OOA) and $O_x$ becomes stronger compared with that during summer. Higher OOA concentration was usually related to both higher $O_x$ concentration and higher RH (Fig. 5g). Meanwhile, the increase trend of OOA concentration with ALWC was also more clear in winter than in summer. These suggest that both photochemical oxidation and aqueous-phase reaction play roles in the OOA formation during winter in Beijing. It should be noted that temperature could affect both atmospheric oxidative capacity and RH and therefore the photochemistry and aqueous-phase processes. As shown in Fig. S6, both mass concentration and mass fraction of OOA showed positive correlations with temperature, suggesting that high temarature promotes the oxidation and formation of OOA in summer. Further analysis shows a positive corralation between temparature and Ox ($R^2$ = 0.59) and a strong negative correlation between temperature and RH ($R^2$ = 0.63). This indicates that high temperature conditions in summer corresponded to high Ox and low RH conditions, further

confirming the relatively important role of photochemical oxidation than aqueous-phase processes in the OOA formation during summer. In winter, however, both mass concentration and mass fraction of OOA (or LO-OOA and MO-OOA) showed no clear correlation with temperature. Also, there was no clear correlation between temperature and Ox or RH, suggesting a more complex effects of temperature on the SOA formation in winter.

The difference in daytime increasing rate of LO-OOA and MO-OOA suggests potential difference in their formation during winter (Fig. 3b). The correlations of the two identified OOAs with $O_x$ and ALWC are investigated in Fig. 6. A clear positive correlation between the mass concentration of LO-OOA and $O_x$ is shown ($R^2$ = 0.51 for the entire data), while the correlation between LO-OOA and ALWC is weaker. Consistently, the correlation coefficient between the mass concentration of LO-OOA and $O_x$ decreased from 0.61 at RH < 60% to 0.29 at RH > 60% (Fig. 6a). The fractional contribution of LO-OOA to total OA increased from 0.17 to 0.35 as $O_x$ concentration increased from 30 ppb to 60 ppb when RH < 60%, mainly driven by the increased LO-OOA concentrations as $O_x$ increases. The fractional contribution of LO-OOA however decreased from 0.25 to 0.18 when ALWC increased from 15 μg m$^{-3}$ to 300 μg m$^{-3}$ (Fig. 6d), explained by the more efficient formation of MO-OOA at high RH conditions during winter. Consistently, the mass concentration of MO-OOA showed a strong positive correlation with ALWC during winter with $R^2$ of 0.6, while the correlations between MO-OOA and $O_x$ are weaker than those of LO-OOA both at RH < 60% and RH > 60%. The fractional contribution of MO-OOA increased clearly from 0.14 to 0.25 when ALWC increased from 60 μg m$^{-3}$ to 350 μg m$^{-3}$, but decreased from 0.3 to 0.18 when $O_x$ concentration increased from 30 ppb to 70 ppb. Similarly, there was a good correlaiton between the ratio of LO-OOA/MO-OOA and $O_x$ concentration when RH < 60% and there was no significant dependence of LO-OOA/MO-OOA on $O_x$ concentration when RH > 60% (Fig. 7). Our results indicate that $O_x$ likely played a more important role in the LO-OOA formation whereas ALWC was more important for the MO-OOA formation in winter. The dependence of LO-OOA as well as the mass ratio of LO-OOA to MO-OOA on atmospheric oxidative tracer (i.e., $O_x$) both degraded when RH were greater than 60%, suggesting that RH or aerosol liquid water may also affect the LO-OOA formation.

## 3.5 Evolution from clean days to pollution days

In order to better understand the aerosol evolution from clean days to pollution days between summer and winter, PM$_1$ composition and OA sources in clean days (daily average PM$_1$ < 20 μg m$^{-3}$) and pollution days (daily average PM$_1$ > 40 μg m$^{-3}$ in summer and daily average PM$_1$ > 60 μg m$^{-3}$ in winter) were analyzed. As shown in Fig. 8, the PM$_1$ composition shows similar evolution trends from clean days to pollution days in both seasons, with increase of SIA contribution and decrease of OA contribution, suggesting the enhanced formation of SIA during pollution period. Specifically, the PM$_1$ mass increased from 13.7 μg m$^{-3}$ during clean days to 59.2 μg m$^{-3}$ during pollution days in summer, with OA fraction decreasing from 60% to 44% and SIA

fraction increasing from 29% to 48%. In comparison, during winter the average mass concentration of PM$_1$ was 115.5 μg m$^{-3}$ during polluted days, which was ten times higher than that during clean days (10.2 μg m$^{-3}$). The mass fraction of OA decreased from 55% during clean days to 48% during pollution days and SIA fraction increased from 36% to 41% accordingly. As for OA, the mass concentrations increased substantially during both summer and winter. In summer, the mass fraction of SOA increased from 70% in clean days to 76% in pollution days along with 3 times greater mass of OA, indicating the importance of SOA formation during pollution events in summer. It should be noted that ISOOA had similar contributions to OA between clean days (5%) and pollution days (5%) with the mass concentration increasing from 0.4 μg m$^{-3}$ to 1.3 μg m$^{-3}$ and the increase in SOA contribution was mainly from OOA. In winter, although the mass fraction of SOA (LO-OOA and MO-OOA) decreased from 44% during clean days to 39% during pollution days, their mass concentrations increased by 8 times. The mass fraction of POA (HOA, COA, BBOA and CCOA) increased from 56% to 61%, indicating that primary contributions are as important as secondary contributions for OA during pollution in winter.

To further investigate the effect of RH on aerosol composition, we separated the pollution days to high-RH pollution days (H-RH, RH > 60%) and low-RH pollution days (L-RH, RH < 40%) in summer and winter. As shown in Fig. 8, the PM$_1$ concentration during L-RH pollution days (60.7 μg m$^{-3}$) was similar to that during H-RH pollution days (58.3 μg m$^{-3}$) during summer. Interestingly, the mass concentration of sulfate decreased dramatically from 18.3 μg m$^{-3}$ during L-RH pollution days to 9.2 μg m$^{-3}$ during H-RH pollution days and the mass fraction decreased accordingly from 30% to 16% in summer. The concentration of nitrate increased from 5.1 μg m$^{-3}$ during L-RH pollution days to 10.9 μg m$^{-3}$ during H-RH pollution days and the mass fraction increased accordingly from 8% to 19%. In contrast, during winter the PM$_1$ concentration during H-RH pollution days (121.4 μg m$^{-3}$) was higher than that during L-RH pollution days (89.5 μg m$^{-3}$). The mass fraction of sulfate increased from 6% during L-RH pollution days to 14% during H-RH pollution days and the mass concentration increased accordingly from 5.8 μg m$^{-3}$ to 16.8 μg m$^{-3}$, while the mass fraction of nitrate decreased from 21% during L-RH pollution days to 16% during H-RH pollution days with similar mass concentration of 18.6 μg m$^{-3}$ and 19.0 μg m$^{-3}$ respectively. The difference of sulfate and nitrate behavior for L-RH and H-RH polluted days in summer and winter suggests that there are factors controlling their production related to aqueous-phase processing other than RH or aerosol liquid water content (e.g., precursors, catalyst, and temperature).

As for SOA, the concentration of OOA decreased from 18.4 μg m$^{-3}$ during L-RH pollution days to 17.0 μg m$^{-3}$ during H-RH pollution days with the mass fraction decreasing from 75% to 68%. Note that the low RH condition is often related to high temperature and strong oxidation capacity, which facilitates photochemical production during summer. As is consistent with previous understanding of different importance of photochemical and aqueous production for the two OOAs, the LO-OOA and MO-OOA in winter vary differently between L-RH pollution days and H-

RH pollution days. The mass fraction of MO-OOA increased from 12% during L-RH pollution days to 17% during H-RH pollution days and the mass concentration increased accordingly from 5.8 µg m$^{-3}$ to 9.9 µg m$^{-3}$. On the contrary, the mass fraction of LO-OOA decreased from 34% during L-RH pollution days to 20% during H-RH pollution days with mass concentration decreasing from 16.1 µg m$^{-3}$ to 11.7 µg m$^{-3}$.

## 4. Conclusion

In this study, variations of PM$_1$ composition, OA sources and secondary formation processes between summer and winter in urban Beijing were analyzed. PM$_1$ composition revealed the increase of SIA contribution and decrease of OA contribution during polluted days when compared to clean days, suggesting the important role of SIA during haze pollution. Higher mass ratio of sulfate to nitrate during L-RH pollution in summer whereas during H-RH pollution in winter were observed. The analysis of RH and O$_x$ effects indicated that photochemical oxidation was probably the major pathway of sulfate production during summer while the importance of aqueous-phase reaction increased during winter. In contrast, much higher nitrate (17%) was found during winter due to preferable gas-to-particle partitioning at low temperature, while higher nitrate (19%) was also observed under the condition of high relative humidity during summer likely due to enhanced gas-to-particle partitioning of hydrophilic NH$_4$NO$_3$ and the hydrolysis of N$_2$O$_5$. As for OA sources, SOA dominated OA mass (74%) during summer while the SOA contribution decreased to 39% during winter due to enhanced primary emission in heating season. OOA had higher concentration and fraction during L-RH pollution days than during H-RH pollution days in summer, indicating the possibility of control factors other than RH or aerosol liquid water for aqueous processing. Summertime OOA was likely dominated by gas-phase photochemical process given the good correlation between OOA and O$_x$. In comparison, MO-OOA increased during H-RH pollution days in winter while the LO-OOA decreased during H-RH pollution days, indicating that the formation of MO-OOA was perhaps promoted by the aqueous-phase processes, while gas-phase photochemical oxidation facilitates the LO-OOA formation during winter. These conclusions were supported by the good correlation between LO-OOA and O$_x$ as well as tight correlation between MO-OOA and ALWC during winter. Meanwhile, we found that the dependence of LO-OOA as well as the mass ratio of LO-OOA to MO-OOA on O$_x$ both degraded under conditions of RH > 60%, suggesting that RH or aerosol liquid water may also affect the LO-OOA formation.

*Data availability.* Raw data used in this study are archived at the Institute of Earth Environment, Chinese Academy of Sciences, and are available on request by contacting the corresponding author.

*Supplement.* The Supplement related to this article is available online at

*Author contributions.* RJH designed the study. Data analysis and source apportionment were done by JD, RJH and QC. JD and RJH wrote the manuscript. JD and RJH interpreted data and prepared display items. All authors commented on and discussed the manuscript.

*Competing interests.* Douglas R. Worsnop is an employee of Aerodyne Research, Inc. (ARI), and an ACSM produced by Aerodyne was used in this study.

*Acknowledgements.* This work was supported by the National Natural Science Foundation of China (NSFC) under grant no. 91644219, no. 41877408 and no. 21661132005, the Chinese Academy of Sciences (no. ZDBS-LY-DQC001), and the National Key Research and Development Program of China (no. 2017YFC0212701). The authors acknowledge financial support from the Cross Innovative Team fund from the State Key Laboratory of Loess and Quaternary Geology (SKLLQG) (no. SKLLQGTD1801). The authors also thank Weikang Ran for assistance in field measurements.

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

**Table 1.** Summary of mass concentrations of aerosol species, gaseous pollutants and meteorological parameters during different periods between summer and winter in Beijing.

| Species | Summer | | | | Winter | | | |
|---|---|---|---|---|---|---|---|---|
| | Entire Study | Clean | L-RH Pollution | H-RH Pollution | Entire Study | Clean | L-RH Pollution | H-RH Pollution |
| Aerosol species ($\mu g\ m^{-3}$) | | | | | | | | |
| $PM_1$ | 41.0 | 13.7 | 60.7 | 58.3 | 63.2 | 10.2 | 89.5 | 121.4 |
| OA | 19.1 | 8.3 | 24.6 | 25.1 | 31.2 | 5.6 | 47.8 | 57.8 |
| HOA | 1.5 | 0.8 | 1.6 | 1.7 | 3.6 | 0.4 | 2.9 | 8.9 |
| COA | 3.4 | 1.7 | 3.1 | 4.7 | 6.3 | 1.4 | 11.3 | 9.6 |
| CCOA | - | - | - | - | 6.1 | 1.0 | 9.1 | 11.6 |
| BBOA | - | - | - | - | 2.8 | 0.3 | 2.6 | 6.0 |
| ISOOA | 1.0 | 0.4 | 1.2 | 1.3 | - | - | - | - |
| OOA | 13.0 | 5.4 | 18.4 | 17.0 | - | - | - | - |
| LO-OOA | - | - | - | - | 7.1 | 1.0 | 16.1 | 11.7 |
| MO-OOA | - | - | - | - | 5.1 | 1.4 | 5.8 | 9.9 |
| $SO_4^{2-}$ | 7.5 | 1.7 | 18.3 | 9.2 | 7.6 | 1.4 | 5.8 | 16.8 |
| $NO_3^-$ | 5.6 | 0.9 | 5.1 | 10.9 | 10.5 | 1.2 | 18.6 | 19.0 |
| $NH_4^+$ | 5.7 | 1.4 | 10.1 | 8.5 | 7.4 | 1.1 | 9.8 | 14.6 |
| $Cl^-$ | 0.2 | 0.1 | 0.2 | 0.3 | 3.1 | 0.3 | 3.9 | 6.2 |
| eBC | 2.9 | 1.3 | 2.4 | 4.3 | 3.4 | 0.6 | 3.6 | 7.0 |
| Gaseous pollutants | | | | | | | | |
| $SO_2$ (ppb) | 3.8 | 3.0 | 6.5 | 3.8 | 10.8 | 4.5 | 14.2 | 16.1 |
| CO (ppm) | 0.6 | 0.4 | 0.7 | 0.6 | 1.8 | 0.5 | 1.7 | 3.7 |
| NO (ppb) | 4.8 | 4.8 | 6.8 | 4.5 | 34.9 | 5.4 | 24.9 | 79.3 |
| $NO_2$ (ppb) | 21.5 | 19.4 | 11.7 | 23.4 | 33.2 | 14.9 | 43.7 | 50.3 |
| $O_3$ (ppb) | 49.6 | 40.0 | 110.2 | 36.1 | 9.2 | 18.6 | 6.3 | 1.2 |
| Meteorological parameters | | | | | | | | |
| RH (%) | 58.1 | 55.9 | 35.7 | 70.4 | 49.3 | 30.3 | 27.8 | 81.9 |
| T (℃) | 28.2 | 26.7 | 32.9 | 26.7 | -2.5 | -2.9 | 0.3 | -3.3 |
| WS ($m\ s^{-1}$) | 0.5 | 0.7 | 0.5 | 0.3 | 1.5 | 2.6 | 1.0 | 0.6 |

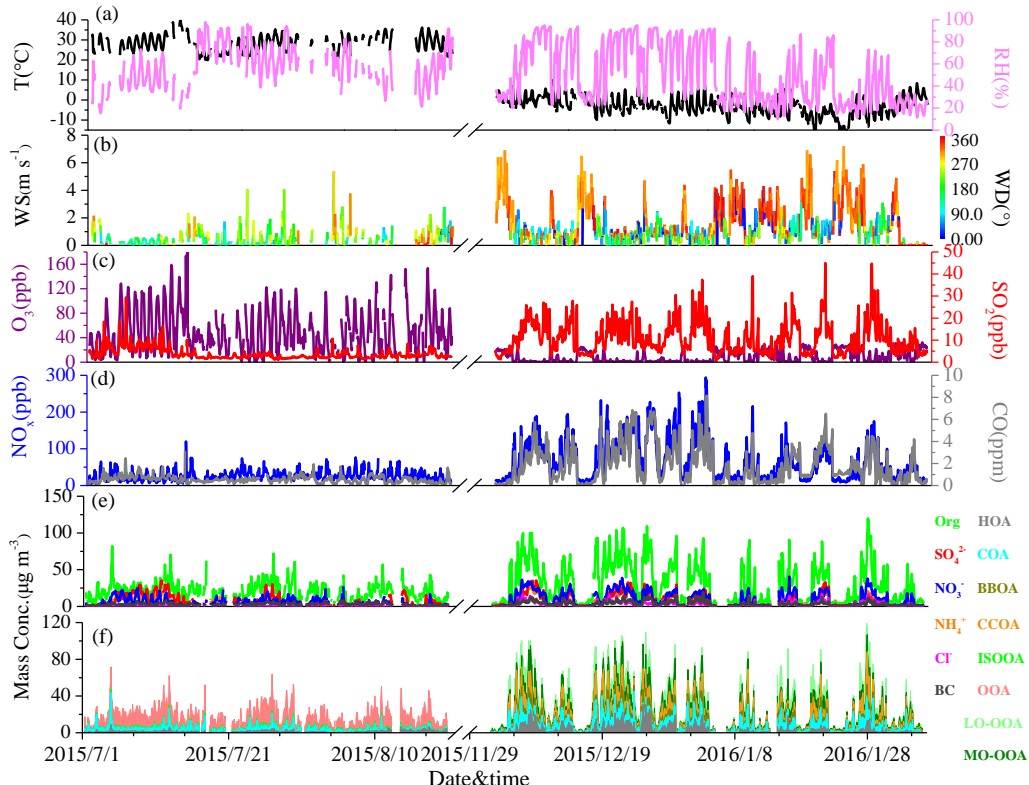

**Figure 1.** Time series of (a) temperature (T) and relative humidity (RH), (b) wind speed (WS) and wind direction (WD), (c) $O_3$ and $SO_2$, (d) CO and $NO_x$, (e) $PM_1$ species, (f) OA sources between summer and winter measurements.

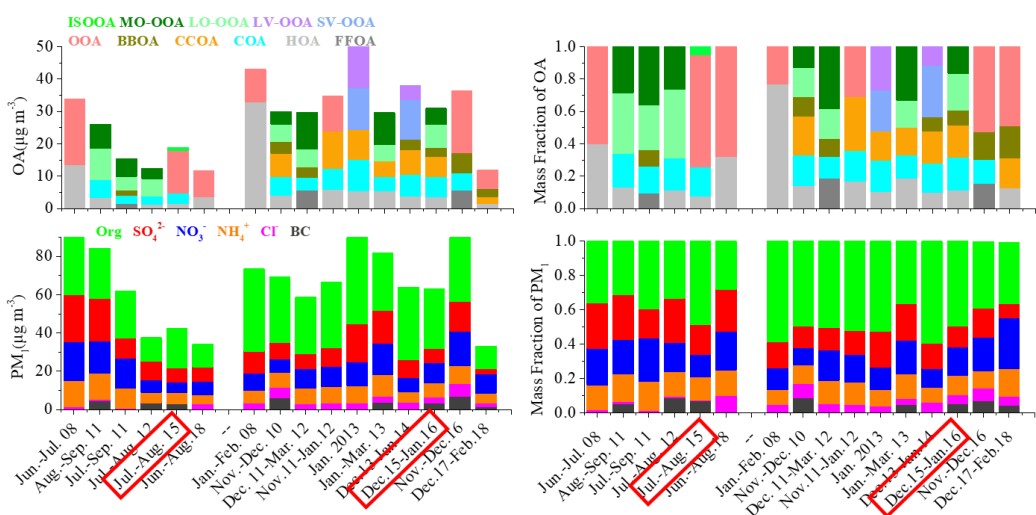

**Figure 2.** Summary of average chemical composition of OA and $PM_1$ in urban Beijing between summer and winter from 2008 to 2018. A more detailed description of the data is presented in Table S1. As for OA sources, fossil fuel related OA(FFOA) is the sum of HOA + CCOA, SV-OOA is semi-volatility OOA and LV-OOA is low-volatility OOA.

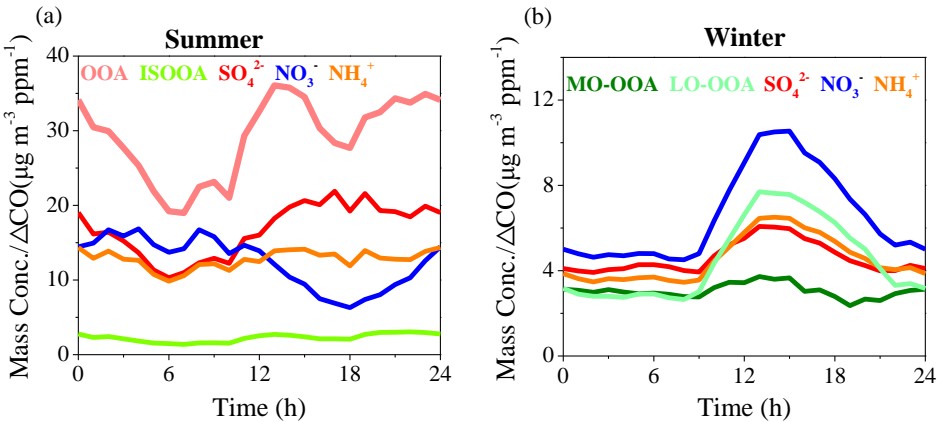

**Figure 3.** Average diurnal variations of secondary species/△CO during summer (a) and winter (b).

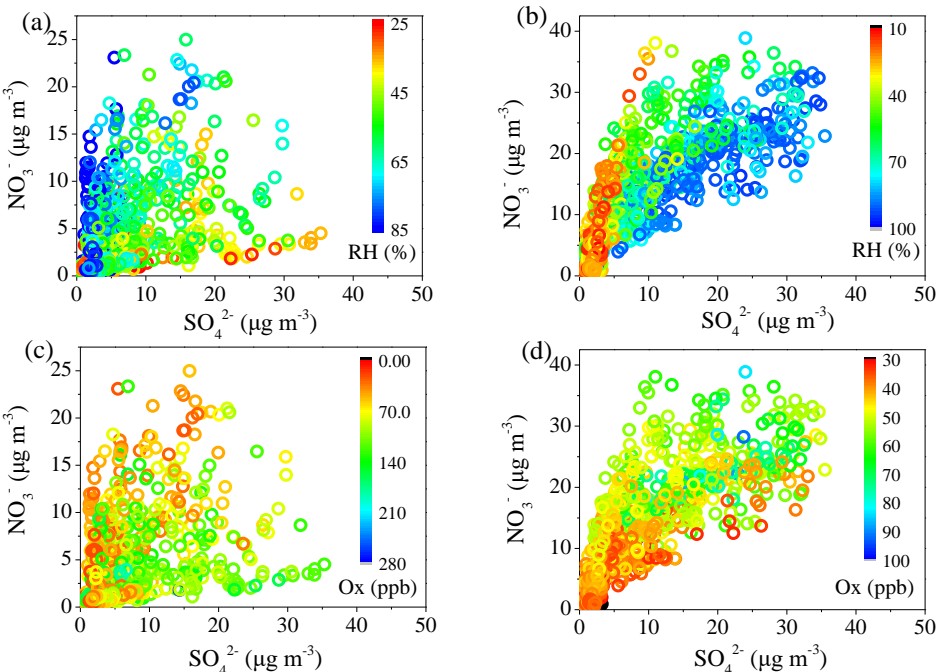

**Figure 4.** Correlations between $NO_3^-$ and $SO_4^{2-}$ color-coded with RH in summer (a) and winter (b), and correlations between $NO_3^-$ and $SO_4^{2-}$ color-coded with $O_x$ concentration in summer (c) and winter (d).

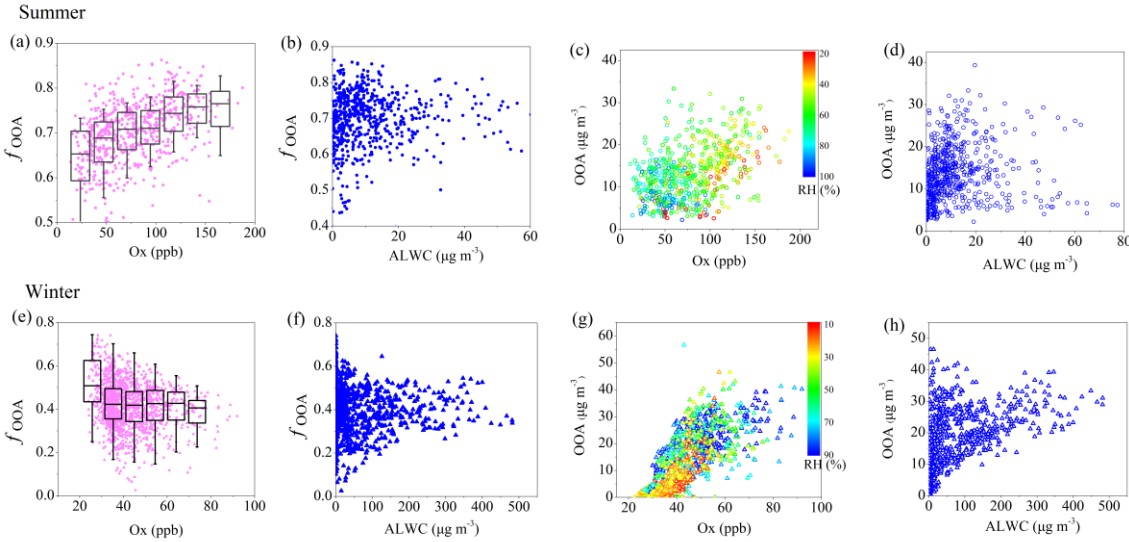

**Figure 5.** Scatterplots of $f_{OOA}$ (mass fraction of OOA (related to OOA in summer and the total of LO-OOA+MO-OOA in winter) to OA) and mass concentration of OOA versus $O_x$ and ALWC in summer (a, b, c, d) and winter (e, f, g, h).

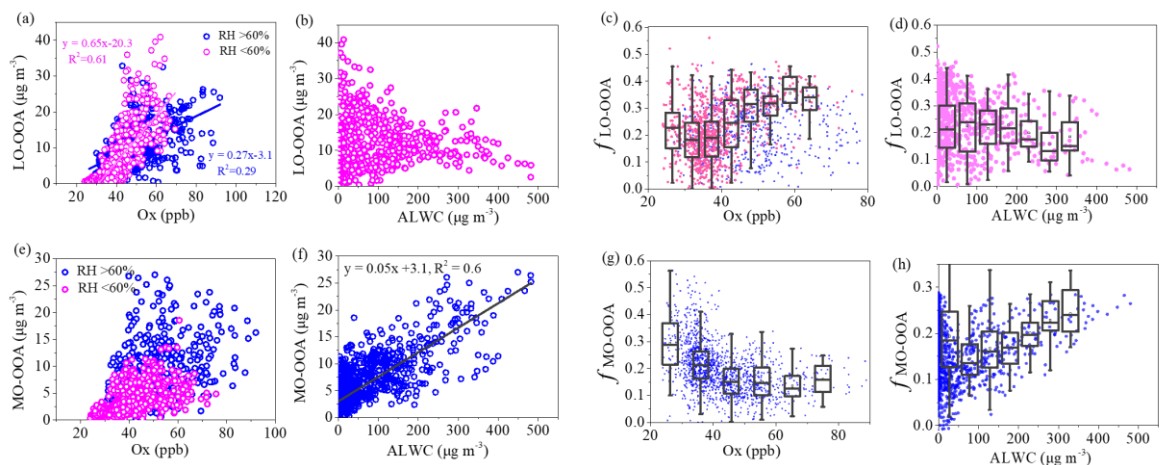

**Figure 6.** Relationship between LO-OOA or MO-OOA and $O_x$ (a, e) and relationship between LO-OOA or MO-OOA and ALWC (b, f), as well as mass fractions of LO-OOA and MO-OOA as functions of Ox (c, g) and ALWC (d, h) in winter.

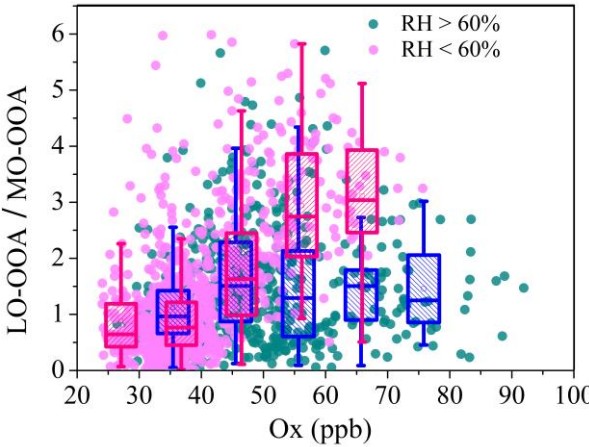

**Figure 7.** Scatterplots of LO-OOA/MO-OOA (mass ratio) versus $O_x$ at RH > 60% and RH < 60% in winter.

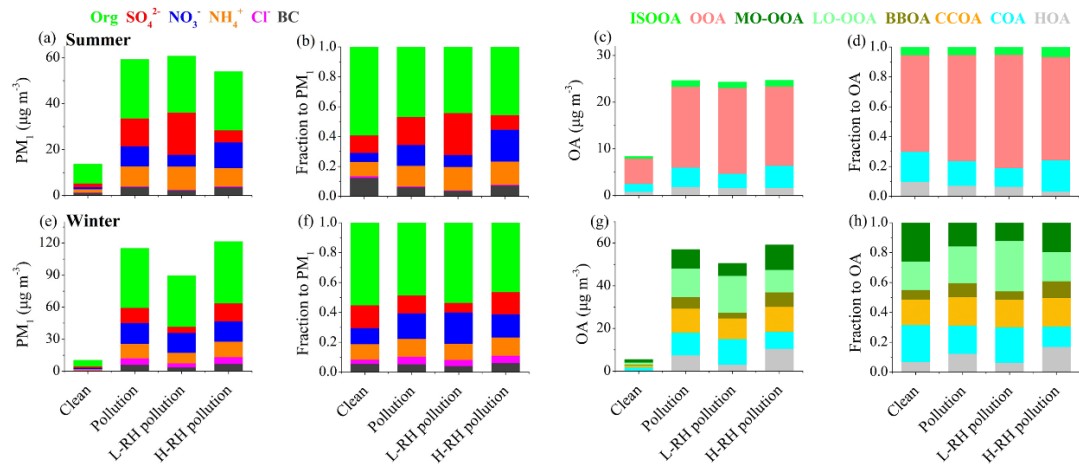

**Figure 8.** Mass concentrations and relative contributions of $PM_1$ species and OA sources during different periods in summer (a, b, c, d) and winter (e, f, g, h).