# Peer review of "Summertime and wintertime atmospheric processes of secondary aerosol in Beijing"

_Atmospheric Chemistry and Physics, 2019_

## Referee Comment (RC1) · Anonymous Referee #2 · 4 Nov 2019

This manuscript reported the composition of submicron aerosol measured by an aerosol chemical speciation monitor (ACSM) in summer and winter in Beijing. PMF analysis is performed for the sources apportionment of organic aerosol. Correlation analysis is used to investigate the formation mechanisms of nitrate, sulfate, and organic aerosol. The same measurements and data analysis have been repeatedly performed in Beijing as well as other megacities in China. This manuscript lacks novelty and in-depth discussions. Most of the conclusions are speculative. I am sorry that I cannot recommend publication in its current format.

Major Comments

1. Page 6 Line 20-23. It is stated that the higher COA concentration in winter than summer suggests enhanced primary emission during winter. However, the changes

are more likely caused by meteorological conditions than emissions, as the cooking activities are not expected to have seasonal variation.

2. Page 6 Line 24-26. In order to evaluate the effects of biomass burning control on air quality, the change in absolute BBOA concentration needs to be used, instead of the fraction of BBOA in total OA.

3. Page 6 Line 33-34. It is a bit surprising that two OOA factors are resolved in winter, but only one OOA factor is resolved in summer. This is different from the observation in Jimenez et al. 2009 Science and many others studies containing summer vs winter contrast. This doesn't necessarily mean that the PMF results in this study are wrong, but it would be interesting to understand why. The instrument resolution is not the only reason, as it doesn't change between seasons.

4. Page 8 Line 12-13. It is an interesting finding that "daytime oxidation formation efficiency of secondary aerosol during winter was comparable to that during summer", but the reason for this observation warrants in-depth analysis. This is just one of many examples that can substantially improve the manuscript.

5. Section 3.3. The rationale behind the scatter plot between NO3 and SO4 is not clear. NO3 and SO4 originates from different precursors, but this difference is ignored in the scatter plot. The conclusions from this analysis are also highly speculative. For example, the larger slope of NO3/SO4 is attributed to that high RH facilitates the gas/particle partitioning of ammonium nitrate. However, wouldn't the high RH (or LWC to be more precise) have similar effect on ammonium sulfate? The formation of nitrate depends on many factors, including NOx, OH, temperature, RH (potentially), etc. The authors need to express more caution in interpreting the results from the simple correlation plots. The conclusions such as RH enhanced nitrate formation in summer and photochemical process dominating nitrate formation in winter are not well supported.

6. Page 9 Line 17-18. It is written that "This suggests that RH has either no effects or complex effects on the OOA formation." This statement only illustrates that the previous

analysis is not meaningful.

7. Section 3.4. Similar to the discussions on the formation mechanism of sulfate and nitrate, the effects of OA precursors are completely omitted in the discussion. Temperature is another important factor when contrasting the OA between summer and winter, but it is also ignored in the discussion.

---

## Referee Comment (RC2) · Anonymous Referee #1 · 14 Nov 2019

The manuscript by Duan et al. reported atmospheric processes of secondary aerosols, including secondary inorganic and secondary organic aerosols in Beijing during summertime and wintertime. Although similar studies have been conducted in Beijing, there are some valuable information presented here. I agree with the reviewer 2 that the authors need to further highlight the novelty of this study in the revised manuscript.

Comments:

1. The authors resolved an isoprene-oxidized OA (ISOOA) in summer by using the constrained profile that was identified in southeastern US. I suggest the authors adding some discussions on the uncertainties in the text. ISOOA was generally formed in environment with low NOx and high biogenic emissions. Because the authors used ME2 for the source apportionment, any input spectral profile can force to separate an OA

factor. 2. The authors claimed that aqueous-phase processed had significant impacts on the formation of sulfate in winter. According to the Figure 3b, the diurnal variations of SO4/CO was remarkably similar to that of NO3/CO. It seems that photochemical production was very important in winter, could the author give more explanations? 3. The color of OA factors in summer is difficult to read (e.g., Figure 1), suggesting the authors to change it.
* * *

---

## Referee Comment (RC3) · Anonymous Referee #3 · 14 Nov 2019

This manuscript by Duan et al. presents measurement results in Beijing summer and winter with a focus on the secondary aerosol formation processes. The paper is overall written in fine English, and present results in a relatively clear way, but there remains some unresolved issues before its acceptance. (1) A natural but critical question is novelty of the findings. The campaigns were conducted in 2015 (a bit old data), and a large number of AMS studies have been conducted in Beijing in recent years. Similar conclusions were already reported previously, for example the formation mechanisms of sulfate (and nitrate) in summer (photochemical dominant) and winter (aqueous-processing dominant); the links between photochemical oxidation with LO-OOA and aq-processing with MO-OOA were also reported previously in Beijing (and some other cities); also, the methods used to conduct such analyses are also similar to prior stud-

ies. The authors must clearly state what new findings this paper can offer, and in the meantime, to highlight the differences of your results from previous ones (for instance, in Fig.2, you should also add results after 2015) (2) P2 Line 33-37. The aqueous formation of sulfate is indeed controversial, therefore I do not suggest the authors to make conclusions on it. For example, you state, "this pathway, has been ruled out. . .". This is still an open question in my opinion. (3) I have a similar doubt with another reviewer regarding the PMF factors. The presence of ISOOA is in question. In urban environment of Beijing, even in summer, it is unlikely to have a significant biogenic SOA factor, this was very likely due to your initial input in ME-2. This is should be carefully checked. In addition, the terminology of different OA factors should be considered more carefully. As you used a Q-ACSM, which is unable to calculate the elemental ratios, how can you define a more (or less) oxidized OOA? If this is only based on the spectral characteristics or similarities with previously identified factors, you should state it clearly in your manuscript. (4) In Figures 4 and 5, it is better to show the correlation coefficients. (5) Some researchers argue that ALWC and Ox may not be effective indicators for aqueous processing and photochemical processing, even though a few studies conducted similar analyses. There could be large uncertainties, and the data are also of large scatter, therefore complicates interpretation of your results. Such uncertainties should be discussed. (6) Other influencing factors should also be considered when you look into the correlation of OOA factors with ALWC or Ox. For example, you tried to minimize the influence of PBL height by using delta CO; such influences from weather conditions rather than chemistry itself may also affect your analyses here. How about you investigate correlation of OOA/delta CO with ALWC, for example? In addition, I also suggest to discuss influences of different air masses on the correlations.

---

## Author Comment (AC1) · 7 Feb 2020

The authors thank the editor and referees to review our manuscript and particularly for the valuable comments and suggestions that are very helpful in improving the manuscript. We provide below point-by-point responses to the referees' comments. We also have made most of the changes suggested by the referees in the revised manuscript.

Please also note the supplement to this comment:
https://www.atmos-chem-phys-discuss.net/acp-2019-910/acp-2019-910-AC1-supplement.pdf

[Figure]

**Supplement:**

The authors thank the editor and referees to review our manuscript and particularly for the valuable comments and suggestions that are very helpful in improving the manuscript. We provide below point-by-point responses to the referees' comments. We also have made most of the changes suggested by the referees in the revised manuscript.

**Referee #1**

The manuscript by Duan et al. reported atmospheric processes of secondary aerosols, including secondary inorganic and secondary organic aerosols in Beijing during summertime and wintertime. Although similar studies have been conducted in Beijing, there are some valuable information presented here. I agree with the reviewer 2 that the authors need to further highlight the novelty of this study in the revised manuscript. Comments:

1. The authors resolved an isoprene-oxidized OA (ISOOA) in summer by using the constrained profile that was identified in southeastern US. I suggest the authors adding some discussions on the uncertainties in the text. ISOOA was generally formed in environment with low NOx and high biogenic emissions. Because the authors used ME2 for the source apportionment, any input spectral profile can force to separate an OA factor.

**Response:** We thank the referee's suggestion. It is indeed necessary to include some uncertainty estimation regarding the ISOOA factor. In our source apportionment using ME-2, the $a$ value approach was used which determines the extent to which the output profiles can differ from the model inputs. We used high $a$ values of 0.3-0.6 to constrain the ISOOA profile, allowing large variation of the ISOOA output profile. To further evaluate the uncertainties of this factor, we performed ME-2 with each $a$ value (0.3-0.6) for 20 times and calculated the standard deviation of these solutions, which is now shown as the error bar in final ISOOA profile (Fig. S3). An uncertainty of ~20% was estimated for ISOOA in our measurement, suggesting a large uncertainty in ISOOA source in urban environment. The low mass fraction and concentration (5% of total OA and an average mass concentration of 1 $\mu g\ m^{-3}$) of this ISOOA factor may be another reason for the relatively high uncertainty.

In the revised supplementary page 2 line 7-14, we have now added the following discussion in source apportionment section:
"To minimize the bias from non-local input profiles, the $a$ value approach was used which determines the extent to which the output profiles can differ from the model inputs. It should be particularly noted that we used high $a$ values of 0.3-0.6 to constrain the ISOOA profile, allowing large variation of the ISOOA output profile. To further evaluate the uncertainties of this factor, we performed ME-2 with each $a$ value (0.3-0.6) for 20 times and calculated the standard deviation of these solutions, which is shown as the error bar in final ISOOA profile (Fig. S3). An uncertainty of ~20% was estimated for ISOOA in our measurement, suggesting a large uncertainty in ISOOA source in urban environment."

In the revised manuscript page 7 line 1-7, we have now added the following discussion:
"…in summertime Beijing. ISOOA was generally thought to be formed in environment with low NOx and high biogenic emissions, but has recently been observed in urban Nanjing in

summer 2013 (Zhang et al., 2017). Similar to the result in summertime Nanjing (4% of OA), in our study ISOOA is found to contribute ~5% of total OA with an average mass concentration of 1.0 μg m$^{-3}$ in summertime Beijing. In contrast, ISOOA was not resolved during winter, which is consistent with the low emissions of isoprene. It should be noted that the estimated uncertainty of ISOOA factor is ~20%, suggesting large uncertainty in ISOOA source in urban Beijing."

2. The authors claimed that aqueous-phase processed had significant impacts on the formation of sulfate in winter. According to the Figure 3b, the diurnal variations of SO4/CO was remarkably similar to that of NO3/CO. It seems that photochemical production was very important in winter, could the author give more explanations?

**Response:** We agree with the reviewer that photochemical oxidation also contributed to wintertime sulfate formation, as shown in Fig. S5h the sulfur oxidation ratio (SOR) in winter still displayed a positive correlation with Ox. However, it should be noted that the increasing rate of SO$_4^{2-}$/△CO was much lower than that of NO$_3^-$/△CO although their diurnal variations were similar. On the other hand, the exponential relationship between SOR and RH was much more obvious in winter than in summer (Fig. S5b and d), and the correlation between SO$_4^{2-}$ and NO$_3^-$ showed that higher concentrations of SO$_4^{2-}$ were related to higher RH conditions in winter. Therefore, we concluded that aqueous-phase processes had relatively significant impacts on the formation of sulfate in winter.

In the revised manuscript page 10 line 11-20, we have now added the following discussion: "It is noted that the diurnal variations of SO$_4^{2-}$/△CO and NO$_3^-$/△CO were similar in winter (Fig. 3). However, the increasing rate of SO$_4^{2-}$/△CO was much lower than that of NO$_3^-$/△CO. Also the absolute concentration of wintertime sulfate showed a decreasing trend in the afternoon, consistent with the diurnal variations of RH and ALWC, while the absolute concentration of wintertime nitrate showed an increasing trend from 8:00 to 18:00 LT (Fig. S4). These results further suggest the difference in dominant formation processes between sulfate and nitrate in winter. The photochemical processes were likely the dominant pathways for daytime nitrate formation but less important for sulfate formation because of large contributions from aqueous-phase processes as indicated by the exponential relationship between SOR and RH in wintertime Beijing (Fig. S5d)."

3. The color of OA factors in summer is difficult to read (e.g., Figure 1), suggesting the authors to change it.

Response: Change made. We have now changed the color of OOA in Figures 1, 2, 3, and 8 in the revised manuscript and Figures S3, S4 in the revised supplementary.

**Referee #2**
This manuscript reported the composition of submicron aerosol measured by an aerosol chemical speciation monitor (ACSM) in summer and winter in Beijing. PMF analysis is

performed for the source apportionment of organic aerosol. Correlation analysis is used to investigate the formation mechanisms of nitrate, sulfate, and organic aerosol. The same measurements and data analysis have been repeatedly performed in Beijing as well as other megacities in China. This manuscript lacks novelty and in-depth discussions. Most of the conclusions are speculative. I am sorry that I cannot recommend publication in its current format.

**Response:** We agree with the referee that an increasing number of ACSM/AMS studies have been conducted in Beijing over the past few years. However, the causes of fine PM pollution in urban Beijing are still not fully understood, most likely due to the campaign-to-campaign difference in meteorological conditions, emissions, and atmospheric processes. In addition, previous studies conducted in different seasons usually focused on sources variations or only SIA or SOA formations (Sun et al., 2015; Hu et al., 2016; Hu et al., 2017; Xu et al., 2017). In our study, we present summer and winter measurements and discuss the seasonal difference in aerosol sources and secondary processes both on SIA and SOA formations. In particular, daytime oxidation formation efficiency of secondary aerosol during winter was found to be comparable to that during summer. Meanwhile, based on the correlation analysis with Ox and ALWC, diurnal variations of secondary species and evolutions from clean to pollution periods with different meteorological conditions between summer and winter, the relative importance of photochemistry versus aqueous-phase processes in the formation of both SIA and SOA were well investigated. Therefore, our study still provides valuable information to the scientific community to improve our understanding of fine PM pollution.

Major Comments
1. Page 6 Line 20-23. It is stated that the higher COA concentration in winter than summer suggests enhanced primary emission during winter. However, the changes are more likely caused by meteorological conditions than emissions, as the cooking activities are not expected to have seasonal variation.

**Response**: We agree with the reviewer. It now reads "COA also showed similar increasing trend from summer to winter, with mass concentration increasing from 3.4 μg m$^{-3}$ to 6.3 μg m$^{-3}$ and fractional contribution from 18% to 20%. The increase in mass concentration of COA in winter is likely caused by meteorological conditions (e.g., shallower PBL heights in winter than in summer) because the cooking activities are not expected to have seasonal variation."

2. Page 6 Line 24-26. In order to evaluate the effects of biomass burning control on air quality, the change in absolute BBOA concentration needs to be used, instead of the fraction of BBOA in total OA.

**Response:** Change made. It now reads "The average wintertime BBOA concentration decreased from 3.6 μg m$^{-3}$ in 2010 (Hu et al., 2016) to 2.8 μg m$^{-3}$ in 2015 in our study and further to 2.4 μg m$^{-3}$ in 2017-2018 (Li et al., 2019), suggesting efficient control of the

biomass burning activities in Beijing and surrounding areas in recent years."

3. Page 6 Line 33-34. It is a bit surprising that two OOA factors are resolved in winter, but only one OOA factor is resolved in summer. This is different from the observation in Jimenez et al. 2009 Science and many others studies containing summer vs winter contrast. This doesn't necessarily mean that the PMF results in this study are wrong, but it would be interesting to understand why. The instrument resolution is not the only reason, as it doesn't change between seasons.

**Response**: This is a good question. In our study, two OOA factors (i.e., LO-OOA and MO-OOA) with much different time series were resolved in winter, and only one OOA factor was resolved in summer. When adding one more factor (i.e., from 4 factors to 5 factors) in the ME-2 model for summertime data, the additional factor is a split from the OOA factor according to the profile and time series (see the figure below). Actually, in some studies summarized in Jimenez et al. (2009), different OOA factors were resolved between summer and winter in the same measurement site using the same instrument. For example, only one OOA factor was resolved in winter and two OOA factors (i.e., SV-OOA and LV-OOA) were resolved in summer for the studies in Zurich (Lanz et al., 2007; 2008) and in Tokyo (Takegawa et al., 2005; 2006). Also the year-long measurement in Beijing conducted by Hu et al. (2017) resolved one OOA factor in spring and autumn, while two OOA factors including MO-OOA and LO-OOA in summer and winter.

[Figure]

Figure 1. Mass spectrums (left) and time series (right) of OA sources for 5 factors solution during summer.

4. Page 8 Line 12-13. It is an interesting finding that "daytime oxidation formation efficiency of secondary aerosol during winter was comparable to that during summer", but the reason for this observation warrants in-depth analysis. This is just one of many examples that can substantially improve the manuscript.

**Response**: We thank the referee's suggestion. We note that wintertime photochemistry was traditionally thought to be low because the wintertime OH radical concentrations were traditionally thought to be low due to low $O_3$ concentrations (e.g., 9.2 ppb in winter versus 49.6 ppb in summer in our study) and reduced solar UV level in winter. However, recent measurements of OH radicals at a suburban site of Beijing reported wintertime OH

radical concentrations at noontime ranged from $2.4\times10^6$ cm$^{-3}$ in severely polluted air ($k_{OH}$ ~27 s$^{-1}$) to $3.6\times10^6$ cm$^{-3}$ in relatively clean air ($k_{OH}$ ~5 s$^{-1}$) (Tan et al., 2018). These OH radical concentrations are nearly one order of magnitude larger than what global models predict for northern China in winter (Lelieveld et al., 2016). The higher-than-expected OH concentrations and moderate OH reactivity in Beijing resulted in fast photochemistry in winter (Lu et al., 2019), explaining the high contribution of secondary aerosol during wintertime severe haze events (Huang et al., 2014).

Regarding the sources of OH radicals, there have been increasing evidence showing HONO, instead of $O_3$, is the main source of OH radicals in wintertime North China Plain. Studies from four years of ground-based observations in urban Beijing and a nearby rural site (Xianghe) show that OH production from HONO photolysis is more than 10 times higher than that from $O_3$ photolysis in winter (Hendrick et al., 2014). Using the TUV radiation model, Xing et al. (2019) shows that in Beijing the estimated net OH production rate from HONO photolysis is $1.0\times10^{-4}$ ppb s$^{-1}$, about 10 time higher than the estimated maximal OH production rate from $O_3$ photolysis ($1.2\times10^{-5}$ ppb s$^{-1}$). Further, Lu et al. (2019) found surprisingly high OH radical oxidation rates of up to 15 ppbv h$^{-1}$ in Beijing in winter, which is comparable to that in summer and is mainly initiated by the photolysis of HONO and maintained by an extremely efficient radical cycling process driven by nitric oxide.

In the revised manuscript from page 8 line 35 to page 9 line 14, we have added the following discussion:
"Certainly, we note that wintertime photochemistry was traditionally thought to be low because the wintertime OH radical concentrations were traditionally thought to be low due to low $O_3$ concentrations (e.g., 9.2 ppb in winter versus 49.6 ppb in summer in our study) and reduced solar UV level in winter. However, recent measurements of OH radicals at a suburban site of Beijing reported wintertime OH radical concentrations at noontime ranged from $2.4\times10^6$ cm$^{-3}$ during severe pollution events ($k_{OH}$ ~27 s$^{-1}$) to $3.6\times10^6$ cm$^{-3}$ during relatively clean events ($k_{OH}$ ~5 s$^{-1}$) (Tan et al., 2018). These OH radical concentrations are nearly one order of magnitude larger than those predicted by global models for northern China in winter (Lelieveld et al., 2016). The higher-than-expected OH concentrations and moderate OH reactivity in Beijing resulted in fast photochemistry in winter (Lu et al., 2019), explaining the high contribution of secondary aerosol during wintertime severe haze events (Huang et al., 2014). Meanwhile, there have been increasing evidence showing HONO, instead of $O_3$, is the main source of OH radicals in wintertime North China Plain (Hendrick et al., 2014; Xing et al., 2019). Further, Lu et al. (2019) found surprisingly high OH radical oxidation rates of up to 15 ppbv h$^{-1}$ in Beijing in winter, which is comparable to that in summer and is mainly initiated by the photolysis of HONO and maintained by an extremely efficient radical cycling process driven by nitric oxide."

5. Section 3.3. The rationale behind the scatter plot between NO3 and SO4 is not clear. NO3 and SO4 originates from different precursors, but this difference is ignored in the scatter plot. The conclusions from this analysis are also highly speculative. For example, the larger slope of NO3/SO4 is attributed to that high RH facilitates the gas/particle partitioning of

ammonium nitrate. However, wouldn't the high RH (or LWC to be more precise) have similar effect on ammonium sulfate? The formation of nitrate depends on many factors, including NOx, OH, temperature, RH (potentially), etc. The authors need to express more caution in interpreting the results from the simple correlation plots. The conclusions such as RH enhanced nitrate formation in summer and photochemical process dominating nitrate formation in winter are not well supported.

**Response**: Combining with the scatter plot between $NO_3^-$ and $SO_4^{2-}$, we also discussed the effects of Ox and RH on SOR and NOR (see Fig. S5), which are related to the sulfate precursor $SO_2$ and nitrate precursor $NO_2$ respectively. Nitrate is a semi-volatile species whose formation can be affected by RH and temperature. High RH facilitates the gas-to-particle partitioning of ammonium nitrate. In comparison, sulfate is non-volatile and will stay in the particle phase. In addition, diurnal variations of $SO_4^{2-}$ and $NO_3^-$ and the evolution from clean days to pollution days with different meteorological conditions were also further used to support the conclusions, such as RH enhanced nitrate formation in summer and photochemical process dominated nitrate formation in winter. With information from these different aspects, we believe that it is a comprehensive analysis and provides valuable information on the formation of sulfate and nitrate.

6. Page 9 Line 17-18. It is written that "This suggests that RH has either no effects or complex effects on the OOA formation." This statement only illustrates that the previous analysis is not meaningful.

**Response:** We were trying to say that in summer photochemical oxidation was the dominant pathway for OOA formation, whereas contributions from aqueous-phase processes were minor and complex. For example, in the condition of low oxidation capacity (low $O_x$), high RH may facilitate the formation of OOA. We have made change and it now reads "This suggests that RH has relatively minor but complex effects on the formation of OOA during summer."

7. Section 3.4. Similar to the discussions on the formation mechanism of sulfate and nitrate, the effects of OA precursors are completely omitted in the discussion. Temperature is another important factor when contrasting the OA between summer and winter, but it is also ignored in the discussion.

**Response**: We agree with the referee that it will be very helpful for in-depth understanding of the SOA formation mechanisms through simultaneous measurements of particulate OA and its precursors (i.e., VOCs). However, in our study we focused on particle phase and did not measured VOCs and therefore could not link SOA with its VOCs. Nevertheless, Liu et al. (2018) measured VOCs in Beijing and found that high concentrations of VOC precursors might contribute to sustained organic aerosol growth and long duration of haze events under typical ambient conditions in Beijing.

Following the referee's suggestion, we have now discussed the effects of temperature on the formation of OOA. As shown in the figure below, both mass concentration and mass

fraction of OOA showed positive correlations with temperature in summer, suggesting that high temperature promotes the oxidation and formation of OOA in summer. Further analysis shows a positive correlation between temperature and Ox ($R^2$ = 0.59) and a strong negative correlation between temperature and RH ($R^2$ = 0.63). This indicates that high temperature conditions in summer corresponded to high Ox and low RH conditions, further confirming the relatively important role of photochemical oxidation than aqueous-phase processes in the OOA formation during summer. In winter, however, both mass concentration and mass fraction of OOA (or LO-OOA and MO-OOA) showed no clear correlation with temperature. Also, there was no clear correlation between temperature and Ox or RH, suggesting a more complex effects of temperature on the SOA formation in winter.

In the revised supplement, we have now added the figure below as Figure S6, and in the revised manuscript page 11 line 2-13, we have added the following discussion:
"It should be noted that temperature could affect both atmospheric oxidative capacity and RH and therefore the photochemistry and aqueous-phase processes. As shown in Fig. S6, both mass concentration and mass fraction of OOA showed positive correlations with temperature, suggesting that high temarature promotes the oxidation and formation of OOA in summer. Further analysis shows a positive corralation between temparature and Ox ($R^2$ = 0.59) and a strong negative correlation between temperature and RH ($R^2$ = 0.63). This indicates that high temperature conditions in summer corresponded to high Ox and low RH conditions, further confirming the relatively important role of photochemical oxidation than aqueous-phase processes in the OOA formation during summer. In winter, however, both mass concentration and mass fraction of OOA (or LO-OOA and MO-OOA) showed no clear correlation with temperature. Also, there was no clear correlation between temperature and Ox or RH, suggesting a more complex effects of temperature on the SOA formation in winter."

[Figure]

Figure 2. Effects of temperature on the formation of OOA in summer (a-d) and winter (e-l). OOA in winter refers to the sum of LO-OOA and MO-OOA.

**Referee #3**

This manuscript by Duan et al. presents measurement results in Beijing summer and winter with a focus on the secondary aerosol formation processes. The paper is overall written in fine English, and present results in a relatively clear way, but there remains some unresolved issues before its acceptance.

(1) A natural but critical question is novelty of the findings. The campaigns were conducted in 2015 (a bit old data), and a large number of AMS studies have been conducted in Beijing in recent years. Similar conclusions were already reported previously, for example the formation mechanisms of sulfate (and nitrate) in summer (photochemical dominant) and winter (aqueous processing dominant); the links between photochemical oxidation with LO-OOA and aq-processing with MO-OOA were also reported previously in Beijing (and some other cities); also, the methods used to conduct such analyses are also similar to prior studies. The authors must clearly state what new findings this paper can offer, and in the meantime, to highlight the differences of your results from previous ones (for instance, in Fig.2, you should also add results after 2015)

**Response**: We agree with the referee that an increasing number of AMS/ACSM studies have been conducted in Beijing over the past few years. However, the causes of fine PM pollution in urban Beijing are still not fully understood, most likely due to the campaign-to-campaign difference in meteorological conditions, emissions, and atmospheric processes. In addition, previous studies conducted in different seasons usually focused on sources variations or only SIA or SOA formations (Sun et al., 2015; Hu et al., 2016; Hu et al., 2017; Xu et al., 2017). In our study, we present summer and winter measurements and discuss the seasonal difference in aerosol sources and secondary processes both on SIA and SOA formations. In particular, we highlight that daytime oxidation formation efficiency of secondary aerosol during winter was comparable to that during summer. Meanwhile, based on the correlation analysis with Ox and ALWC, diurnal variations of secondary species and evolutions from clean to pollution periods with different meteorological conditions between summer and winter, the relative importance of photochemistry versus aqueous-phase processes in the formation of both SIA and SOA were well investigated. Therefore, our study still provides valuable information to the scientific community to improve our understanding of fine PM pollution.

Following the referee's suggestion, we have added results in Beijing after 2015 in the revised Fig. 2. In page 7 line 23-26, we have added "From summer 2015 to summer 2018, the PM$_1$ concentration continued decreasing while the SIA contribution was again higher than that during summer 2015 and the SOA faction was similar to that in summer 2015 (Zhou et al., 2019)."

In page 7 line 31-35, we have added "After 2015, the wintertime PM$_1$ concentration further decreased to 33.2 $\mu$g m$^{-3}$ in 2017-2018 (Li et al., 2019), but showed a peak concentration (92.9 $\mu$g m$^{-3}$) in Nov-Dec 2016 because of prevailing air masses from south and northeast of Beijing and high frequencies of high RH and low wind speeds (Xu et al., 2018). Such changes again suggested that the difference in meteorological conditions is one of the major causes of severe particulate pollution in Beijing." and in line 38, we added

"However, from winter 2015 to 2018, the SIA fraction increased from 40% to 54%, suggesting the increased importance of SIA from 2015 to 2018."

(2) P2 Line 33-37. The aqueous formation of sulfate is indeed controversial, therefore I do not suggest the authors to make conclusions on it. For example, you state, "this pathway, has been ruled out*: : :*". This is still an open question in my opinion.

**Response**: Change made. It now reads "Moreover, the aqueous oxidation of $SO_2$ by $NO_2$ has been suggested to be an efficient pathway for sulfate formation (Cheng et al., 2016; Wang et al., 2016), although contribution from this pathway in real air is controversial and is still an open question (Guo et al., 2017; Liu et al., 2017)."

(3) I have a similar doubt with another reviewer regarding the PMF factors. The presence of ISOOA is in question. In urban environment of Beijing, even in summer, it is unlikely to have a significant biogenic SOA factor, this was very likely due to your initial input in ME-2. This is should be carefully checked. In addition, the terminology of different OA factors should be considered more carefully. As you used a Q-ACSM, which is unable to calculate the elemental ratios, how can you define a more (or less) oxidized OOA? If this is only based on the spectral characteristics or similarities with previously identified factors, you should state it clearly in your manuscript.

**Response**: We agree with the referee that biogenic SOA has minor contribution to OA in urban environment of Beijing. In our study, ISOOA contributed to only 5% of total OA (or average mass concentration of 1 μg m$^{-3}$). In our source apportionment using ME-2, the *a* value approach was used which determines the extent to which the output profiles can differ from the model inputs. We used high *a* values of 0.3-0.6 to constrain the ISOOA profile, allowing large variation of the ISOOA output profile. To further evaluate the uncertainties of this factor, we performed ME-2 with each *a* value (0.3-0.6) for 20 times and calculated the standard deviation of these solutions, which is now shown as the error bar in final ISOOA profile (Fig. S3). An uncertainty of ~20% was estimated for ISOOA in our measurement, suggesting a large uncertainty in ISOOA source in urban environment.

As for LO-OOA and MO-OOA factors, although no elemental ratios could be resolved in Q-ACSM, the difference in ratio of intensity at $m/z$ 44 over that at $m/z$ 43 ($f_{44/43}$) can be used to differentiate the oxidation state of OOA for UMR data (Ng et al., 2010). This method has been used in many ACSM studies (e.g., Sun et al., 2016, 2018; Wang et al., 2017; Xu et al., 2017). In our study, as shown in the supplement, two OOA factors with much different time series were resolved, and we defined one as more oxidized OOA (MO-OOA) and the other as less oxidized OOA(LO-OOA) because of much higher $f_{44/43}$ value of MO-OOA (8.6) than that of LO-OOA (2.6).

In the revised supplement page 2 line 7-14, we have now added the following discussion in source apportionment section:
"To minimize the bias from non-local input profiles, the *a* value approach was used which determines the extent to which the output profiles can differ from the model inputs. It

should be particularly noted that we used high *a* values of 0.3-0.6 to constrain the ISOOA profile, allowing large variation of the ISOOA output profile. To further evaluate the uncertainties of this factor, we performed ME-2 with each *a* value (0.3-0.6) for 20 times and calculated the standard deviation of these solutions, which is shown as the error bar in final ISOOA profile (Fig. S3). An uncertainty of ~20% was estimated for ISOOA in our measurement, suggesting a large uncertainty in ISOOA source in urban environment."

In the revised manuscript page 7 line 1-7, we have now added the following discussion: "...in summertime Beijing. ISOOA was generally thought to be formed in environment with low NOx and high biogenic emissions, but has recently been observed in urban Nanjing in summer 2013 (Zhang et al., 2017). Similar to the result in summertime Nanjing (4% of OA), in our study ISOOA is found to contribute ~5% of total OA with an average mass concentration of 1.0 μg m$^{-3}$ in summertime Beijing. In contrast, ISOOA was not resolved during winter, which is consistent with the low emissions of isoprene. It should be noted that the estimated uncertainty of ISOOA factor is ~20%, suggesting large uncertainty in ISOOA source in urban Beijing."

In page 5 line 17-19, we have now added "Note that MO-OOA and LO-OOA were defined in winter because of much higher $f_{44/43}$ value of MO-OOA (8.6) than that of LO-OOA (2.6) and the large difference in time series of these two factors."

(4) In Figures 4 and 5, it is better to show the correlation coefficients.

**Response**: Thanks for the suggestion. In Fig. 4, we investigated the effects of RH and Ox on the relationship between $SO_4^{2-}$ and $NO_3^-$ and thus did not show the correlation coefficients. Actually, there was a good correlation between $SO_4^{2-}$ and $NO_3^-$ in winter with $R^2$ value of 0.67, while the correlation between $SO_4^{2-}$ and $NO_3^-$ was weak for all data in summer ($R^2$=0.13). In Fig. 5, the data points are rather scattering and thus we used box plots to see the trends for some figures. Generally, these two figures show the general trends and thus we did not show the correlation coefficients.

(5) Some researchers argue that ALWC and Ox may not be effective indicators for aqueous processing and photochemical processing, even though a few studies conducted similar analyses. There could be large uncertainties, and the data are also of large scatter, therefore complicates interpretation of your results. Such uncertainties should be discussed.

**Response**: We agree with the referee that there are uncertainties when using ALWC and Ox as indicator to investigate the aqueous-phase processes and photochemical processes. Therefore, more studies and sophisticated instruments are required to find out more effective indicators for aqueous-phase and photochemical processes of SOA. For example, the measurements of OH radicals to replace Ox could be more effective for investigating the atmospheric oxidative capacity. We have added the following uncertainty statement as a caution in the revised manuscript page 10 line 22-25:
"It should be note that a majority of the data are scattering, therefore caution should be

given for the uncertainties when using $O_x$ and ALWC as indicators of photochemical processing and aqueous processing, respectively."

(6) Other influencing factors should also be considered when you look into the correlation of OOA factors with ALWC or Ox. For example, you tried to minimize the influence of PBL height by using delta CO; such influences from weather conditions rather than chemistry itself may also affect your analyses here. How about you investigate correlation of OOA/delta CO with ALWC, for example? In addition, I also suggest to discuss influences of different air masses on the correlations.

**Response**: We agree with the referee that the formation of OOA is affected by many factors including chemical, physical and meteorological factors. The complexity in OOA formation and the campaign-to-campaign difference are the main motivation of our study. When investigating the effects of Ox and ALWC on OOA formation, we have actually shown both absolute mass concentration and mass fraction of OOA (see Fig. 5 and 6). The mass fraction of OOA ($f_{OOA}$, $f_{LO\text{-}OOA}$, or $f_{MO\text{-}OOA}$) is actually a value normalized to total OA, and therefore can minimize the influence of PBL (similar to those normalized to delta CO). Following the referee's suggestion, we have further investigated the influences of different air masses on the correlations (see the figures below). The cluster analysis results show 5 clusters of air masses in summer and 3 clusters of air masses in winter. However, there were no significant differences in these correlations for different air masses in both summer and wither.

[Figure]

Figure 3. Influences of different air masses on the correlations between SO4 and NO3, as well as OOA and Ox or ALWC during summer and winter.

References:

Cheng, Y. F., Zheng, G. J., Wei, C., Mu, Q., Zheng, B., Wang, Z. B., Gao, M., Zhang, Q., He, K. B., Carmichael, G., Pöschl, U., and Su, H.: Reactive nitrogen chemistry in aerosol water as a source of sulfate during haze events in China, Science Advances, 2(12), e1601530, https://doi.org/10.1126/sciadv.1601530, 2016.

Guo, H., Weber, R. J., Nenes, A.: High levels of ammonia do not raise fine particle pH sufficiently to yield nitrogen oxide-dominated sulfate production, Sci. Rep., 7(1), 12109, 2017.

Hendrick, F., Müller, J. F., Clémer, K., Wang, P., De Mazière, M., Fayt, C., Gielen, C., Hermans, C., Ma, J. Z., Pinardi, G., Stavrakou, T., Vlemmix, T., and Van Roozendael, M.: Four years of ground-based MAX-DOAS observations of HONO and NO2 in the Beijing area, Atmos.

Chem. Phys., 14, 765–781, 2014.

Hu, W., Hu, M., Hu, W., Jimenez, J. L., Yuan, B., Chen, W., Wang, M., Wu, Y., Chen, C., Wang, Z., Peng, J., Zeng, L., and Shao, M.: Chemical composition, sources, and aging process of submicron aerosols in Beijing: Contrast between summer and winter, J. Geophys. Res. Atmos., 121(4), 1955–1977, https://doi.org/10.1002/2015JD024020, 2016.

Hu, W., Hu, M., Hu, W.-W., Zheng, J., Chen, C., Wu, Y., and Guo, S.: Seasonal variations in high time-resolved chemical compositions, sources, and evolution of atmospheric submicron aerosols in the megacity Beijing, Atmos. Chem. Phys., 17, 9979–10000, https://doi.org/10.5194/acp-17-9979-2017, 2017.

Huang, R. J., Zhang, Y. L., Bozzetti, C., Ho, K. F., Cao, J. J., Han, Y. M., Daellenbach, K. R., Slowik, J. G., Platt, S. M., Canonaco, F., Zotter, P., Wolf, R., Pieber, S. M., Bruns, E. A., Crippa, M., Ciarelli, G., Piazzalunga, A., Schwikowski, M., Abbaszade, G., Schnelle-Kreis, J., Zimmermann, R., An, Z., Szidat, S., Baltensperger, U., Haddad, I.E., and Prevot, A.S.H.: High secondary aerosol contribution to particulate pollution during haze events in China, Nature, 514, 218–222, 2014.

Jimenez, J. L., Canagaratna, M. R., Donahue, N. M., Prevot, A. S. H., Zhang, Q., Kroll, J. H., DeCarlo, P. F., Allan, J. D., Coe, H., Ng, N. L., Aiken, A. C., Docherty, K. S., Ulbrich, I. M., Grieshop, A. P., Robinson, A. L., Duplissy, J., Smith, J. D., Wilson, K. R., Lanz, V. A., Hueglin, C., Sun, Y. L., Tian, J., Laaksonen, A., Raatikainen, T., Rautiainen, J., Vaattovaara, P., Ehn, M., Kulmala, M., Tomlinson, J. M., Collins, D. R., Cubison, M. J., E, Dunlea, J., Huffman, J. A., Onasch, T. B., Alfarra, M. R., Williams, P. I., Bower, K., Kondo, Y., Schneider, J., Drewnick, F., Borrmann, S., Weimer, S., Demerjian, K., Salcedo, D., Cottrell, L., Griffin, R., Takami, A., Miyoshi, T., Hatakeyama, S., Shimono, A., Sun, J. Y., Zhang, Y. M., Dzepina, K., Kimmel, J. R., Sueper, D., Jayne, J. T., Herndon, S. C., Trimborn, A. M., Williams, L. R., Wood, E. C., Middlebrook, A. M., Kolb, C. E., Baltensperger, U., and Worsnop, D. R.: Evolution of organic aerosols in the atmosphere, Science, 326, 1525–1529, doi:10.1126/science.1180353, 2009.

Lanz, V. A., Alfarra, M. R., Baltensperger, U., Buchmann, B., Hueglin, C., and Prévôt, A. S. H.: Source apportionment of submicron organic aerosols at an urban site by factor analytical modelling of aerosol mass spectra, Atmos. Chem. Phys., 7, 1503–1522, https://doi.org/10.5194/acp-7-1503-2007, 2007.

Lanz, V. A., Alfarra, M. R., Baltensperger, U., Buchmann, B., Hueglin, C., Szidat, S., Wehrli, M. N., Wacker, L., Weimer, S., Caseiro, A., Puxbaum, H., and Prévôt, A. S. H.: Source attribution of submicron organic aerosols during wintertime inversions by advanced factor analysis of aerosol mass spectra, Environ. Sci. Technol., 42(1), 214-220,2008.

Lelieveld, J., Gromov, S., Pozzer, A., and Taraborrelli, D.: Global tropospheric hydroxyl distribution, budget and reactivity, Atmos. Chem. Phys., 16, 12477–12493, https://doi.org/10.5194/acp-16-12477-2016, 2016.

Li, H., Cheng, J., Zhang, Q., Zheng, B., Zhang, Y., Zheng, G., and He, K.: Rapid transition in winter aerosol composition in Beijing from 2014 to 2017: response to clean air actions, Atmos. Chem. Phys., 19, 11485–11499, https://doi.org/10.5194/acp-19-11485-2019, 2019.

Liu, J., Chu, B., Chen, T., Liu, C., Wang, L., Bao, X., and He, H.: Secondary organic aerosol formation from ambient air at an urban site in Beijing: effects of OH exposure and precursor concentrations. Environ. Sci. Technol., 52(12), 6834-6841, 2018.

Liu, M., Song, Y., Zhou, T., Xu, Z., Yan, C., Zheng, M., Wu, Z., Hu, M., Wu, Y., and Zhu, T.: Fine particle pH during severe haze episodes in northern China, Geophys. Res. Lett., 44(10), 5213-5221, 2017.

Lu, K., Fuchs, H., Hofzumahaus, A., Tan, Z., Wang, H., Zhang, L., Schmitt, S., Rohrer, F., Bohn, B., Broch, S., Dong, H., Gkatzelis, G., Hohaus, T., Holland, F., Li, X., Liu, Y., Liu, Y. H., Ma, X., Novell, A., Schlag, P., Shao, M., Wu, Y., Wu, Z., Zeng, L., Hu, M., Kiendler-Scharr, A., Wahner, A., and Zhang, Y.: Fast Photochemistry in Wintertime Haze: Consequences for Pollution Mitigation Strategies, Environ. Sci. Technol., 53(18), 10676-10684, 2019.

Ng, N. L., Canagaratna, M. R., Zhang, Q., Jimenez, J. L., Tian, J., Ulbrich, I. M., Kroll, J. H., Docherty, K. S., Chhabra, P. S., Bahreini, R., Murphy, S. M., Seinfeld, J. H., Hildebrandt, L., Donahue, N. M., DeCarlo, P. F., Lanz, V. A., Prévôt, A. S. H., Dinar, E., Rudich, Y., and Worsnop, D. R.: Organic aerosol components observed in Northern Hemispheric datasets from Aerosol Mass Spectrometry, Atmos. Chem. Phys., 10, 4625–4641, https://doi.org/10.5194/acp-10-4625-2010, 2010.

Sun, Y. L., Wang, Z. F., Du, W., Zhang, Q., Wang, Q. Q., Fu, P. Q., Pan, X., Li, J., Jayne, J., and Worsnop, D. R.: Long-term real-time measurements of aerosol particle composition in Beijing, China: seasonal variations, meteorological effects, and source analysis, Atmos. Chem. Phys., 15, 10149–10165, https://doi.org/10.5194/acp-15-10149-2015, 2015.

Sun, Y., Chen, C., Zhang, Y., Xu, W., Zhou, L., Cheng, X., Zheng, H., Ji, D., Li, J., Tang, X., Fu, P., and Wang, Z.: Rapid formation and evolution of an extreme haze episode in Northern China during winter 2015, Sci. Rep., 6(1), 1-9, 2016.

Sun, Y., Xu, W., Zhang, Q., Jiang, Q., Canonaco, F., Prévôt, A. S., Fu, P., Li, J., Jayne, J., Worsnop, D. R., and Wang, Z.: Source apportionment of organic aerosol from two-year highly time-resolved measurements by an aerosol chemical speciation monitor in Beijing, China, Atmos. Chem. Phys., 18(12), 8469–8489, 2018.

Takegawa, N., Miyazaki, Y., Kondo, Y., Komazaki, Y., Miyakawa, T., Jimenez, J. L., Jayne, J. T., Worsnop, D. R., Allan, J. D., and Weber, R. J.: Characterization of an Aerodyne Aerosol Mass Spectrometer (AMS): Intercomparison with other aerosol instruments, Aerosol Sci. Tech., 39(8), 760-770, 2005.

Takegawa, N., Miyakawa, T., Kondo, Y., Jimenez, J. L., Zhang, Q., Worsnop, D. R., and Fukuda, M.: Seasonal and diurnal variations of submicron organic aerosol in Tokyo observed

using the Aerodyne aerosol mass spectrometer, J. Geophys. Res. Atmos., 111(D11), 2006.

Tan, Z., Rohrer, F., Lu, K., Ma, X., Bohn, B., Broch, S., Dong, H., Fuchs, H., Gkatzelis, G. I., Hofzumahaus, A., Holland, F., Li, X., Liu, Y., Liu, Y., Novelli, A., Shao, M., Wang, H., Wu, Y., Zeng, L., Hu, M., Kiendler-Scharr, A., Wahner, A., and Zhang, Y.: Wintertime photochemistry in Beijing: observations of ROx radical concentrations in the North China Plain during the BEST-ONE campaign, Atmos. Chem. Phys., 18, 12391–12411, https://doi.org/10.5194/acp-18-12391-2018, 2018.

Wang, G. H., Zhang, R. Y., Gomez, M. E., Yang, L. X., Zamora, M. L., Hu, M., Lin, Y., Peng, J. F., Guo, S., Meng, J. J., Li, J. J., Cheng, C. L., Hu, T. F., Ren, Y. Q., Wang, Y. S., Gao, J., Cao, J. J., An, Z. S., Zhou, W. J., Li, G. H., Wang, J. Y., Tian, P. F., Marrero-Ortiz, W., Secrest, J., Du, Z. F., Zheng, J., Shang, D. J., Zeng, L. M., Shao, M., Wang, W. G., Huang, Y., Wang, Y., Zhu, Y. J., Li, Y. X., Hu, J. X., Pan, B., Cai, L., Cheng, Y. T., Ji, Y. M., Zhang, F., Rosenfeld, D., Liss, P. S., Duce, R. A., Kolb, C. E., and Molina, M. J.: Persistent sulfate formation from London Fog to Chinese haze, Proc. Natl. Acad. Sci. U.S.A., 113(48), 13630–13635, 2016.

Wang, Y. C., Huang, R. J., Ni, H. Y., Chen, Y., Wang, Q. Y., Li, G. H., Tie, X. X., Shen, Z. X., Huang, Y., Liu, S. X., Dong, W. M., Xue, P., Fröhlich, R., Canonaco, F., Elser, M., Daellenbach, K.R., Bozzetti, C., Haddad, EI., and Cao, J. J.: Chemical composition, sources and secondary processes of aerosols in Baoji city of northwest China, Atmos. Environ., 158, 128–137, https://doi.org/10.1016/j.atmosenv.2017.03.026, 2017.

Xing, L., Wu, J., Elser, M., Tong, S., Liu, S., Li, X., Liu, L., Cao, J., Zhou, J., El-Haddad, I., Huang, R., Ge, M., Tie, X., Prévôt, A. S. H., and Li, G.: Wintertime secondary organic aerosol formation in Beijing–Tianjin–Hebei (BTH): contributions of HONO sources and heterogeneous reactions, Atmos. Chem. Phys., 19, 2343–2359, https://doi.org/10.5194/acp-19-2343-2019, 2019.

Xu, W. Q., Han, T. T., Du, W., Wang, Q. Q., Chen, C., Zhao, J., Zhang, Y. J., Li, J., Fu, P. Q., Wang, Z. F., Worsnop, D. R., and Sun, Y. L.: Effects of Aqueous-Phase and Photochemical Processing on Secondary Organic Aerosol Formation and Evolution in Beijing, China, Environ. Sci. Technol., 51(2), 762–770, https://doi.org/10.1021/acs.est.6b04498, 2017.

Xu, W. Q., Sun, Y. L., Wang, Q. Q., Zhao, J., Wang, J. F., Ge, X. L., Xie, C. H., Zhou, W., Du, W., Li, J., Fu, P. Q., Wang, Z. F., Worsnop, D. R., and Coe, H.: Changes in aerosol chemistry from 2014 to 2016 in winter in Beijing: insights from high resolution aerosol mass spectrometry, J. Geophys. Res. Atmos., 124(2), 1132–1147, 2018.

Zhang, Y. J., Tang, L. L., Sun, Y. L., Favez, O., Canonaco, F., Albinet, A., Couvidat, F., Liu, D. T., Jayne, J. T., Wang, Z., Croteau, P. L., Canagaratna, M. R., Zhou, H. C., Prevot, A. S. H., and Worsnop, D.R.: Limited formation of isoprene epoxydiols-derived secondary organic aerosol under NOx-rich environments in Eastern China, Geophys. Res. Lett., 44(4),

2035–2043, https://doi.org/10.1002/2016GL072368, 2017.

Zhou, W., Gao, M., He, Y., Wang, Q., Xie, C., Xu, W., Zhao, J., Du, W., Qiu, Y., Lei, L., Fu, P., Wang, Z., Worsnop, D.R., Zhang, Q., and Sun, Y.: Response of aerosol chemistry to clean air action in Beijing, China: Insights from two-year ACSM measurements and model simulations, Environ. Pollut., 255, 113345, 2019.